# Human brain prefrontal cortex proteomics identifies compromised energy metabolism and neuronal function in Schizophrenia

Frank Koopmans [1], Anke A. Dijkstra[2,3], Wei-Ping Li [1,4], Remco V. Klaassen[1], Yvonne Gouwenberg[1], Shuyang Yao [5], Lisa Bast [6], Matthijs Verhage[7,8], Robert Karlsson [5], Andrew J. Dwork[9], Craig A. Stockmeier [10], Jens Hjerling-Leffler [6], Patrick F. Sullivan [5,11] & August B. Smit [1] ✉

Understanding the pathophysiological substrates of schizophrenia is a major challenge for current neuropsychiatric research. As part of a set of multi-omics experiments, we performed an extensive case-control proteomics study on 192 post-mortem tissue sections from prefrontal cortex from 96 individuals, including 47 cases with schizophrenia and 49 healthy controls. Using two independently measured cortical datasets, we identified 387 proteins differentially expressed between schizophrenia cases and controls at a 5% FDR threshold. This significantly regulated set of proteins contains genes located in GWAS-identified schizophrenia loci and proteins identified by pQTL analysis. Gene ontology analysis using GOAT provided evidence for regulation of several major protein categories, emphasizing downregulation of mitochondrial oxidative respiration, ribosomes and the proteasome, upregulation of kinases and (small) GTPases. SynGO analysis supports the notion of synaptic dysfunction in schizophrenia, with major regulators of pre- and postsynaptic function compromised. Our findings highlight the complex molecular dysregulation in schizophrenia, with mitochondrial function downregulated versus signaling and trafficking upregulated, and synapse function disrupted; in combination with prior avenues of research, these finding support a role for energy deficits compromising highly ATP dependent neuronal function as a target for therapeutic interventions.

Schizophrenia is a complex and often disabling disorder with multifactorial origins, influenced by a combination of genetic, psychological, and environmental factors[1,2]. It is associated with an increased risk of suicide and serious mental and physical health conditions, leading to reduced life expectancy and substantial health and socio-economic costs. Understanding the pathophysiological substrates of schizophrenia remains a major challenge in neuropsychiatric research[1,3].

The onset of schizophrenia is usually in adolescence and early adulthood, a critical period of dynamic brain maturation. Key neurodevelopmental processes may involve alterations in glutamatergic synapses[4,5], maturation of interneurons[6,7], dopaminergic neurotransmission[8], and developmental myelination[9]. Despite extensive studies on various functional and structural brain alterations associated with schizophrenia, the molecular mechanisms underlying neuronal and cognitive dysfunctions remain largely elusive. Findings from multiple studies over the last decades have implicated the prefrontal cortex as playing a central role in the pathophysiology of schizophrenia, specifically linking structural change to cognitive impairments characteristic of the disorder[10,11].

Genetic studies, including genome-wide association studies (GWAS) of schizophrenia have identified nearly 300 common susceptibility loci[12], as well as rare copy number variants[13] and rare coding variants[14]. However, the specific causal variants and their biological consequences remain largely unknown[1].

In recent years, transcriptome and proteome studies have been conducted to uncover the disorder's molecular architecture[15–20]. Despite these advances, omics studies of post-mortem brain tissue samples have lagged behind genetic studies due to individual expression variability in the human brain, the subtle effects of schizophrenia on protein expression, and the lack of power due to the limited sample sizes, all of which contribute to small effect sizes of regulated proteins. Additionally, confounding factors such as different medications and substances used in individual patients present additional challenges. To specifically address these issues, we: (I) utilized a relatively large cohort size and statistically analyzing the independently measured cortical Layers 1–3 and 4–6, while interpreting outcome of both; (II) carefully analyzed the data structure with respect to potential confounders; (III) employed the GOAT algorithm for analysis of Gene Ontology (GO) gene sets.

In this study, we leveraged advancements in mass spectrometry sensitivity and throughput to analyze the dorsal prefrontal cortex (DPFC) proteome of a well-characterized cohort of 96 individuals, comprising 47 cases with schizophrenia and 49 controls, using data-independent acquisition. Notably, our findings revealed significant top-regulated proteins specifically involved in synaptic signaling, endo- and exocytosis, the cytoskeleton, and intracellular transport, several of which were found enriched in previous studies of schizophrenia. Refined enrichment analysis of GO terms identified, amongst others, alterations in the mitochondrial oxidative phosphorylation pathway and in a broad category of protein kinases and small GTPase signaling proteins.

## Results

### Proteomics of Layers 1–3 and Layers 4–6 of dorsal prefrontal cortex from 47 schizophrenia patients and 49 controls

Post-mortem dorsal prefrontal cortex (DPFC) tissue was cryo-dissected using LCM from 47 clinically well-defined schizophrenia (SCZ) cases and 49 controls (CON). For all samples, clinical metadata, including post-mortem delay time of autopsy, age, and sex, were gathered (Supplementary Data 1). Control donors were closely age-matched to SCZ patients, with median ages of 50 and 47.5 years old, respectively (Figure S1). Alzheimer's disease pathology (i.e., amyloid β deposits, neurofibrillary tangles, and neuritic plaques) was staged according to the ABC criteria[21–23], and donor tissues with regional signs of protein aggregation were excluded from further analysis.

Frozen sections of the DPFC from all donors were counterstained with Toluidine Blue to visualize layer 4 and white matter boundaries. When Layer 4 could not be identified using Toluidine blue, SMI-32 staining on nearby sections was used to reveal the border of Layer 3 (Figure S2). Sections were subsequently dissected into Layers 1–3 and 4–6. The dissected tissue samples were processed in a block-randomized order and analysed by mass spectrometry (Fig. 1a). In-depth data analysis and quality control was performed using MS-DAP, identifying in total 36,176 peptides using criteria for consistent detection throughout the samples (Methods), consolidated into 6,293 reliably identified protein groups that mapped to 5,243 unique proteins across experimental conditions.

The distributions of coverage across all individual samples were consistent (Figure S3a), with highly reproducible quantitative peptide and protein data, resulting in a low coefficient of variation (23.3–25.3%) between samples (Figure S3b,c). Differential expression analysis of cortical Layers of the same individuals showed substantial layer-specific differences in protein expression with 3,173 of 5,243 (60.5%) proteins in control samples differentially expressed between Layers

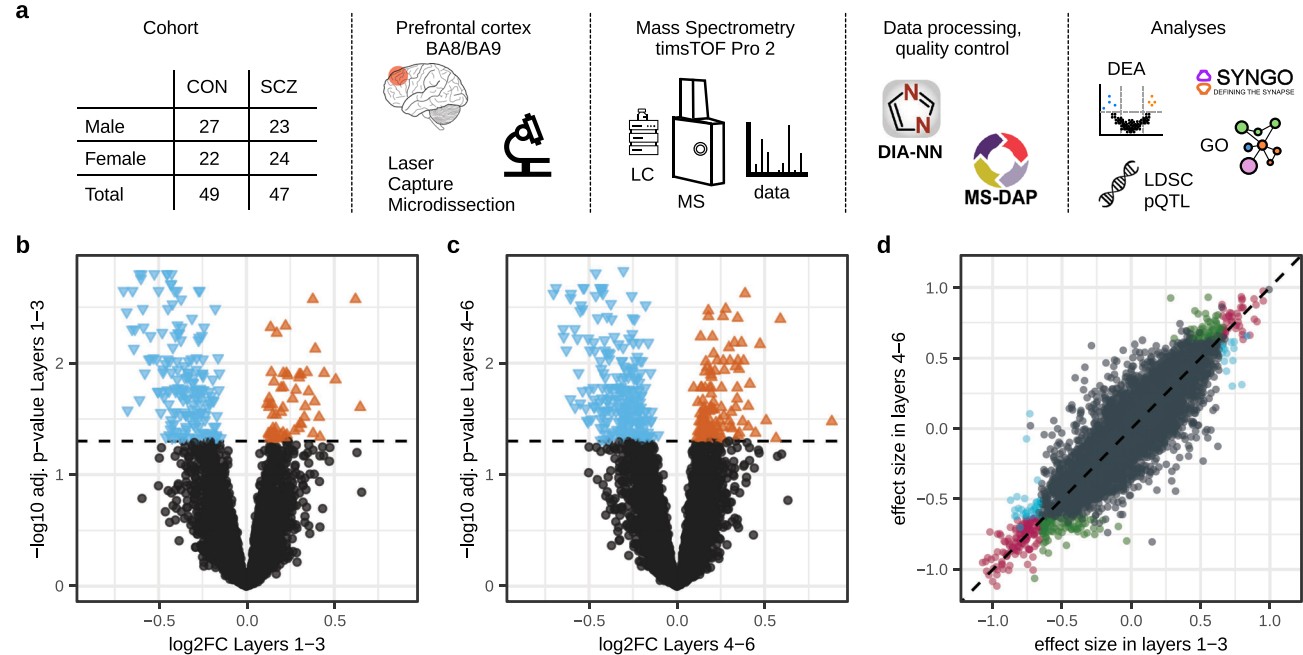

**Fig. 1 | Overview of the study design and proteomics results. a** Schematic representation of the workflow of the study. **b, c** Independent differential expression analyses were performed using limma and DEqMS (Methods) to compare control and schizophrenia cases within each set of cortical Layers. Differentially expressed proteins (FDR adjusted *p*-value < 0.05) in schizophrenia cases as compared to controls are highlighted in blue (triangle pointing down) and orange (triangle pointing up) for lower and higher expression in schizophrenia cases, respectively. **d** Protein effect sizes of control versus schizophrenia in Layers 1–3 from (**b**) (x-axis) and Layers 4–6 from (**c**) (y-axis) were strongly correlated (Pearson's rho 0.88 for all proteins, 0.97 for proteins significant in either Layer set), suggesting that protein dysregulation in schizophrenia was consistent throughout cortical Layers 1–3 or Layers 4–6. Proteins are color-coded when significant (at FDR adjusted *p*-value < 0.05) only in Layers 1-3 (blue), only in Layers 4–6 (green) or in both Layer sets (red). The diagonal is denoted by a dashed line. Source data are provided as a Source Data file.

1–3 and 4–6 (adjusted $p$-value < 0.05; Supplementary Data 2). Typically, Layer 4-6 samples contained some myelin-derived proteins, which may be expected due to the partial intermingling with white matter and Layer 6 in the crown of the gyrus. Independent statistical testing of controls (CON) versus schizophrenia (SCZ) cases identified 229 and 338 significantly altered proteins (adjusted $p$-value < 0.05) in Layers 1–3 and 4–6, respectively, resulting in 387 unique differentially expressed proteins overall (Fig. 1b, c, Supplementary Data 2). Bootstrapping procedures were used to estimate the empirical null distribution of protein effect sizes that would be obtained if sample labels (i.e., assigned phenotype) were randomly assigned (Figure S4). We found that even the smallest effect sizes from significant hits in our CON vs SCZ analyses were outliers in the empirical null, confirming that the reported significant hits are highly unlikely to result from an overinterpretation of noisy data. Protein dysregulation observed in schizophrenia cases was strongly correlated between Layers 1–3 and 4–6 (Pearson's rho 0.97 for proteins significant in either Layer set and 0.88 for all proteins; Fig. 1d).

## No confounding properties in the protein expression data

A crucial aspect of any omics-based analysis of post-mortem tissue is the potential confounding effect arising from technological processes (e.g., sample handling, batch effects), intrinsic biological variables (e.g., age), and external influences (e.g., medication, substance use). Potential confounders were therefore addressed using PCA for patient-relevant parameters, including age of death (Figure S5A), post-mortem delay (PMD) (Figure S5b, h), sex (Figure S5c), and age (Figure S5h), as well as for several technical steps (Figure S5f,g) and sample source (Figure S5e). We found that only the sources of samples (cohorts) contributed to the outcome of the primary data. To also account for minor confounding factors that might not be detected in PCA, we incorporated all relevant covariates observed in PCA or MS-DAP QC analyses (source, gel ID, PMD, age at death) into the linear regression model used to test for differences between cases with schizophrenia and controls (Fig. 1b, c, Methods).

To assess the potential confounding effects of medications and licit drug use, we classified the brain donors in the Stockmeier cohort ($N = 56$), being the largest cohort with detailed medication records, according to alcohol use, nicotine use, and clinical drug treatment. The proteomic profiles of schizophrenia cases were compared between substance-exposed conditions. Case counts with information on alcohol use 28 (19 no, 9 yes), nicotine dependence to 29 (12 no, 16 yes), and drug treatment to 29 (6 no, 23 yes), respectively. Within the boundaries of available case counts, no statistically significant proteins after multiple testing correction for any of the three properties were found. Correlations between the respective substance-exposed effect sizes (Figure S6) and the control vs SCZ effect sizes identified were close to zero ($R^2$ values for alcohol use 0.0008, nicotine 0.144, and 1e-6 for medication treatment). Additionally, unsupervised cluster analysis of the protein abundance matrix showed clustering by the source of samples, with no apparent clustering by other covariates nor did we observe clustering of SCZ patients into subgroups (Figure S7).

Taken together, the absence of statistically significant proteins for all tested confounder properties and their low correlation with effect sizes for control vs schizophrenia cases, supports that significant differential expression and their over-represented pathways represent genuine dysregulation in schizophrenia.

## Differential expression analysis

Proteins lower expressed in schizophrenia cases (Fig. 1b, c, Supplementary Data 2) showed top ranking for the *COX5A* mitochondrial protein. Additional notable proteins in the down-regulated list are calmodulin 1 (*CALM1*, $\log_2$ FC −0.60 in Layers 1–3, −0.55 in Layers 4–6) and calmodulin like 3 (*CALML3*, $\log_2$ FC −0.65 in both Layer sets), known from calcium-dependent intracellular signalling, along with

other calcium-sensing proteins such as *SYT12* ($\log_2$ FC −0.18 in both Layer sets). Clathrin proteins *CLTA* (log2 FC −0.43, −0.38) and *CLTB* ($\log_2$ FC −0.61, −0.55) known from the endocytosis pathway, and tropomyosin 3 (*TPM3*, $\log_2$ FC −0.49 in Layers 4-6) acting together with *CAPZA1* ($\log_2$ FC −0.47, −0.48) in actin-mediated bulk endocytosis[24]. The complexins 1 (*CPLX1*, $\log_2$ FC −0.49, −0.69) and −2 (*CPLX2*, $\log_2$ FC −0.52, −0.53) are proteins involved in synaptic vesicle exocytosis, regulated next to other regulated synaptic proteins. Large groups of down-regulated proteins, as part of cellular organelles, include those of the inner membrane of the mitochondrion, the ribosome, and the proteasome.

The top up-regulated proteins in schizophrenia cases (Fig. 1b, c, Supplementary Data 2) include the previously schizophrenia-linked protein *S100B* ($\log_2$ FC 0.62, 0.59)[25]. Both *S100B* and *COX5A* (down-regulated) were identified in previous post-mortem schizophrenia studies[25,26] and validate our proteomic analysis. *MAPK3* ($\log_2$ FC 0.38, 0.39) belongs to a significantly up-regulated group of kinases, among which are *MAPK1, PTK2B, PRKCG, PRKDC, GRK2, WNK4, RPS6KA1/2, CAMK2A/B/G*. We also found a set of motor proteins *KIF1A* ($\log_2$ FC 0.16 in both Layer sets) and *KIF2A* ($\log_2$ FC 0.15, 0.16), extracellular matrix proteins collagen *COL6A1* ($\log_2$ FC 0.39, 0.45) and *COL6A2* ($\log_2$ FC 0.47, 0.51). Clathrin heavy chain *CLTC* ($\log_2$ FC 0.18, 0.21) was up-regulated, like *DNM1L* ($\log_2$ FC 0.44, 0.47) which is involved in clathrin-mediated endocytosis, oppositely of clathrin light chain proteins *CLTA/B*. The cholinergic receptor *CHRNA4* was differentially expressed at $\log_2$ FC 0.47 in both Layer sets, albeit just beyond the significance threshold at adjusted $p$-values of 0.076 and 0.064.

## pQTL analysis

Protein quantitative trait loci (pQTL) are genetic variants that affect the quantity of that particular protein. Our pQTL analysis (Methods) resulted in identifying 20 significant (FDR adjusted $p$-value < 0.05) proteins (Fig. 2b, Supplementary Data 3). The protein *ANXA11*, a calcium-dependent phospholipid binding protein, was also identified as significantly differentially expressed protein in cases vs controls (Fig. 1b, c, Supplementary Data 2). Functional annotation using the GO database revealed that 8/20 significant pQTL proteins are annotated against the cellular component term mitochondrion (*FAM210B, KYAT3, MMAB, P2RX7, RHOT2, UQCR10, VARS2, VWA8*), while 15/20 significant pQTL proteins are annotated against the biological process term metabolic process (*ABHD17C, BTN2A1, CALCOCO1, CBR1, GALC, HLA-E, KYAT3, LDAH, MKKS, MMAB, P2RX7, PON2, SRR, UQCR10, VARS2*). This corroborates a major down-regulation of mitochondrial proteins that was also observed in our differential expression analysis (Supplementary Data 2).

## Regulated proteins are enriched for SNP-association in schizophrenia and bipolar I disorder

To identify whether the top regulated proteins are supported by a genetic signal, we performed a stratified linkage disequilibrium score analysis (S-LDSC) for different genetic traits on GWAS results for schizophrenia[12], bipolar disorder type-1 (BiP)[27], major depressive disorder (MDD)[28], autism spectrum disorder (ASD)[29] and IQ[30]. The top 200 up- and down-regulated proteins (Fig. 2a) were significantly enriched for SNP-heritability (FDR adjusted $p$-value < 0.05) only for schizophrenia and BiP (Fig. 2c, Supplementary Data 4). The point estimates of the enrichment fold were higher for down-regulated proteins than the up-regulated proteins. The S-LDSR adds evidence that the significantly dysregulated proteins in the schizophrenia post-mortem human brain are enriched in the most recent GWAS data and that SCZ and BiP likely have shared pathology.

## Gene set enrichment analysis

In addition to the top-regulated proteins, we examined whether molecular pathways exhibited differential co-expression in cases vs

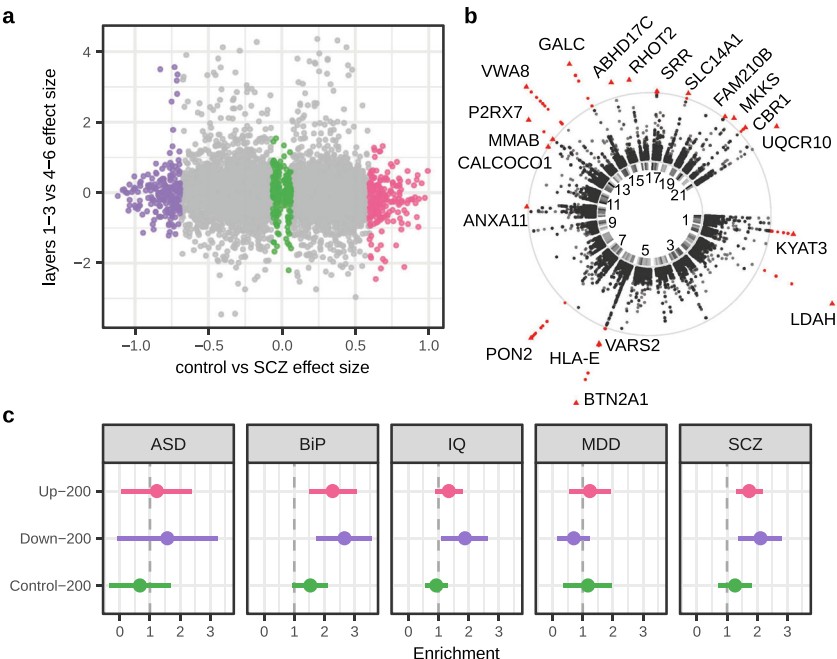

**Fig. 2 | MS analysis of DPFC Layers of schizophrenia cases (SCZ) and controls and its genetic load. a** Summary of differential protein expression for diagnosis (SCZ case/control) vs differences between cortical Layers. From analyses across both Layer sets (Fig. 1b, c, Supplementary Data 2), the highest absolute effect size of each protein (across Layer sets) is shown on x-axis and the respective effect size between Layer sets, for controls only, is shown on the y-axis (Methods). Color-coding indicates the top/center/bottom-200 proteins identified in control vs SCZ; pink: up-regulated in cases; purple: down-regulated in cases, green: 200 proteins with lowest effect size. **b** pQTL analysis applied to the protein abundance matrix yields 376 significant SNPs at FDR adjusted *p*-value < 0.05 (in red, other tested SNPs are shown in black) across 20 genes. Gene labels are shown for their respective most-significant SNP (triangle symbol). The center ring denotes chromosomes 1-22, with labels shown for odd numbers. **c** A stratified LDSC analysis for different genetic traits was run on GWAS data from Autism spectrum disorder, Bipolar disorder type 1, IQ, Major Depressive Disorder and Schizophrenia. Three sets of 200 proteins (each) from the control vs SCZ analysis highlighted in (**a**) were subjected to enrichment testing for each genetic trait; schizophrenia down-regulated (purple), up-regulated (pink) proteins and as a control 200 proteins with lowest effect size (green). From the 5 GWAS tested both schizophrenia and Bipolar show significant protein enrichment at 5% FDR, whereas the top ranked proteins do not reach significance for MDD, ASD and IQ. Data are presented as the S-LDSC enrichment statistic (dot), with error bars indicating the 95% confidence interval. Source data are provided as a Source Data file.

controls. We applied our GOAT algorithm[31] for gene set enrichment analysis to the Gene Ontology (GO) database (Methods) and identified 272 and 303 GO terms in Layers 1-3 and 4-6, respectively, resulting in a set of 338 unique GO terms across both layer sets (Fig. 3, Supplementary Data 5). Consistent with the strong correlation observed for protein effect sizes between Layer sets (Fig. 1d), GO term enrichment scores were highly correlated between Layer sets (Pearson's rho 0.97, Fig. 3d). For down-regulated proteins, mitochondrion-associated GO terms were enriched with proteins from the GO terms specifically localized to the inner (but not outer) mitochondrial membrane (Fig. 3a–c), thus likely impacting on mitochondrial function. This finding is in line with the snRNA-seq data of Bast et al. using samples from the same cohort[32]. The down-regulated mitochondrial inner membrane proteins are part of complexes I, II, IV and V, performing oxidative phosphorylation and ATP production. Complex IV includes the top down-regulated protein *COX5A*. Complex V subunit *ATP5F1C* is another top-hit among down-regulated proteins (ranks 3 and 5 in Layers 1-3 and 4-6, respectively), which reflects the ontology terms related to ATP synthesis and proton-transporting ATP synthase complexes. Key proteins in complex II, connecting with the TCA cycle (e.g., *SDHA/B*) were also down-regulated. Moreover, proteins from mitochondrial and cytoplasmatic ribosomes show lower expression (Fig. 3e). Related to this, we also found a general down-regulation for proteins associated with the more general GO term RNA processing, ATP biosynthetic processes/metabolism, and the proteasome (Fig. 3c, e). Interestingly, some protein phosphatases are also down-regulated in contrast to kinases (Fig. 3c, e).

For up-regulated proteins, the GO-term enrichment analysis indicated a large group of kinases. These kinases include both the serine/threonine as well as the tyrosine kinases (Supplementary Data 5). A large GO term related to small ATPases was impacted, and proteins annotated in GO as 'GTPase activator' or 'GTPase regulator' are mostly found among significantly up-regulated proteins (Fig. 3c, e). In fact, the molecular functions 'ATP binding' (694 proteins, of which 24 and 48 were significantly different in Layers 1-3 and 4-6, respectively) and 'ATP-dependent activity' (281 proteins, of which 16 and 26 were significantly different in Layers 1-3 and 4-6, respectively) have generally higher expression in schizophrenia cases. Finally, proteins involved in membrane trafficking (ER-Golgi-endosomal route) and 'response to oxygen-containing compound' were up-regulated.

Our GOAT analyses of the GO database yielded no significant enrichment for synaptic ontology terms beyond 'synaptic vesicle', even though several of the top differentially expressed proteins are synaptic proteins. To further investigate synaptic involvement, we analysed the schizophrenia differential expression results using the SynGO[33] database (release 1.2 containing 1602 unique synaptic proteins). In total, 1213 of 5243 proteins that were evaluated in control vs schizophrenia were present in the SynGO database, including 122 of the 387 proteins that were significant in at least one cortical Layer set. The SynGO webtool identified significant enrichment for cellular components of both the presynapse and postsynapse, specifically the presynaptic endocytic zone, the postsynaptic cytoskeleton and postsynaptic ribosome (the latter reflecting ribosomal proteins in general rather than synapse-specific proteins). Gene count representation revealed a prominent group of 28 out of 122 proteins/ genes localized

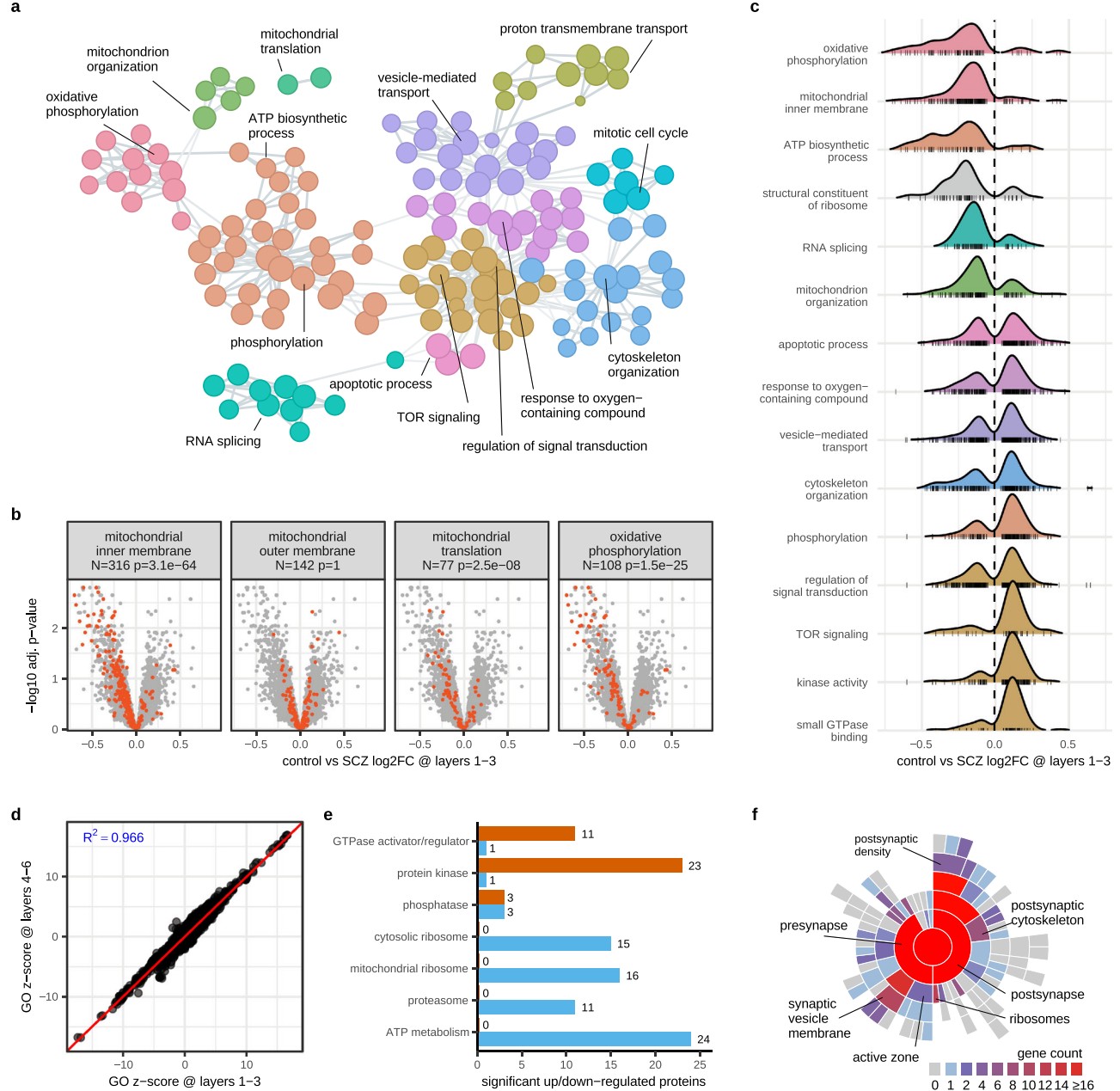

**Fig. 3 | Gene set enrichment analyses for schizophrenia (SCZ) proteins.**
**a** Biological processes identified in Layers 1-3 or 4-6 (or both) by GOAT are visualized as 153 nodes in a network, with edges representing gene overlap. Nodes are colored according to unsupervised clustering. Representative terms are labeled for each cluster. **b** Volcano plots show control vs schizophrenia differential expression statistics in Layers 1–3, i.e. same data as shown in Fig. 1b, with constituents from various mitochondria-related gene sets highlighted in red to illustrate the enrichment trends. The number of identified constituents for each gene set is indicated by 'N' and 'p' indicates the FDR adjusted p-value from GOAT statistical analysis. **c** SCZ log2 foldchange distributions (data from Fig. 1b) of proteins from selected gene sets reveal that some biological processes are strongly enriched in either up- or down-regulated proteins. Colors match the respective clusters in (**a**). **d** GO term enrichment z-scores for both Layer sets are highly correlated. **e** Number of significant proteins (from Fig. 1b, c) identified in Layers 1-3 or 4-6 (or both) that are up- (orange) or down-regulated (blue) within various molecular functions and cellular components. Similar to (**c**), we identify various major protein classes that are near-exclusively up- or down-regulated. **f** Sunburst plot depicting SynGO analysis reveals pre- and post-synaptic proteins among significant hits identified in Layers 1-3 or 4-6 (or both). Source data are provided as a Source Data file.

to the postsynaptic density, including *CAMK2A/B, HOMER1, GRK2, DLGAP1,* and transsynaptic protein *NLGN4*[12] (Fig. 3f, Supplementary Data 6).

### High convergence with single nucleus RNA sequencing pathway enrichments
We compared the significant GO terms identified in our proteomics study to results from the accompanying study by Bast et al.[32] which

applied snRNA-seq to a subset of 42 control and 41 schizophrenia samples of our cohort. 130 out of 338 gene sets across GO domains CC, BP and MF that were significantly enriched in our proteomics dataset were also significant in at least one of the 16 tested cell-types in the snRNA-seq study (adjusted *p*-value < 0.05). Overlapping gene sets were identified across multiple cell-types, with most overlap observed in excitatory neurons and astrocytes (Fig. 4a, Supplementary Data 5). Considering the set of GO terms that are highly significant in snRNA-

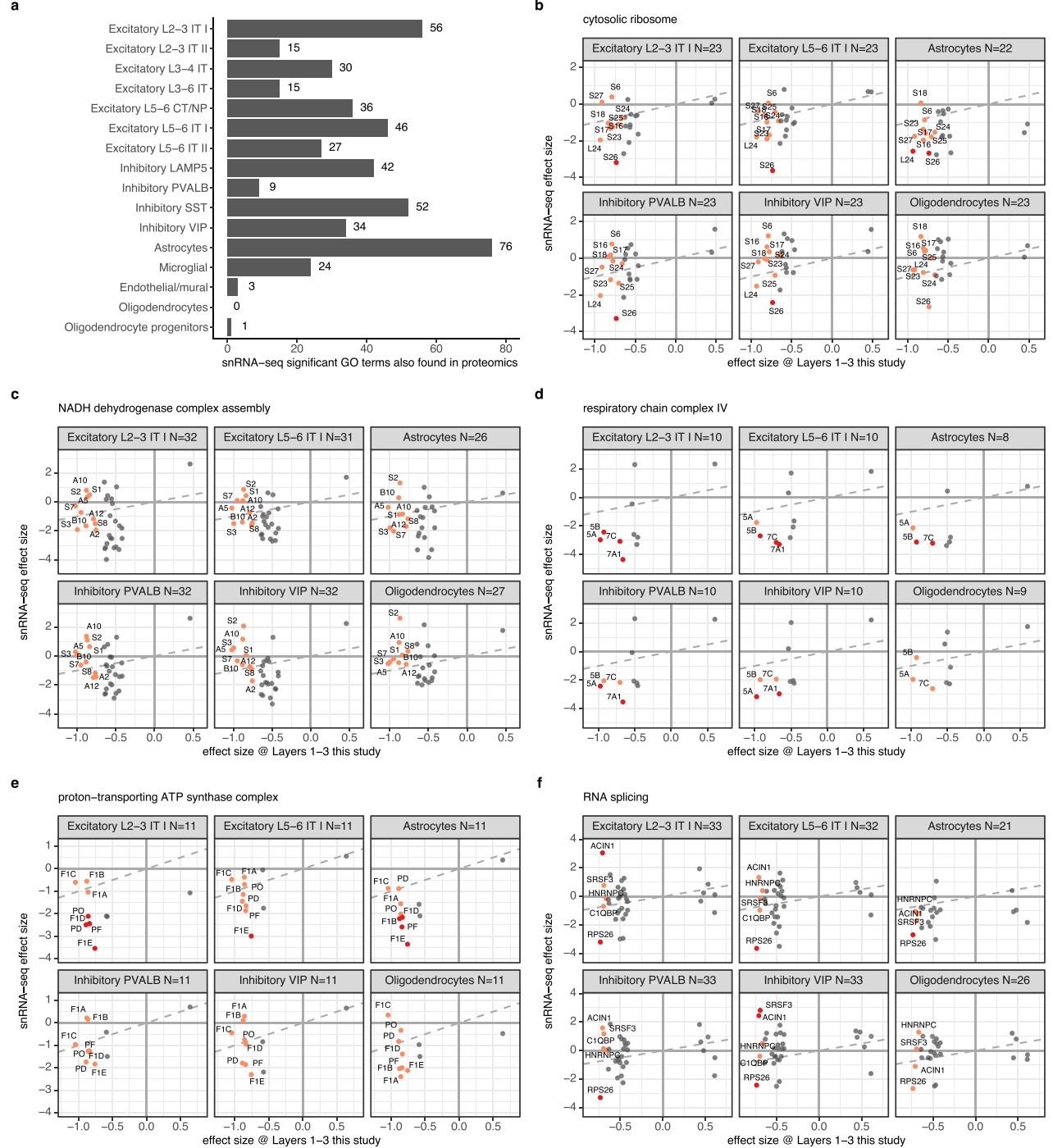

**Fig. 4 | Comparison of schizophrenia (SCZ) associated proteins and pathways between proteomics and snRNA-seq. a** For 338 significant GO terms identified across Layer sets in the proteomics dataset (Fig. 3), the overlap with significant GO terms identified in respective cell-types in the accompanying snRNA-seq study was counted. **b–f** Comparison between protein effect sizes observed in the proteomics control vs SCZ analysis of Layers 1-3 presented in Fig. 1b (Supplementary Data 2) on the x-axis and respective DESeq2 effect sizes observed in the accompanying snRNA-seq study by Bast et al. on the y-axis for 5 selected gene sets (plot title). The number of proteins in each plot is indicated by 'N' in the title. Gene symbols were shortened for visual clarity as follows, (**b**); 'RPS' and 'RPL' were replaced by 'S' and 'L', **c**; 'NDUF' was removed from labels, (**d**) 'COX' was removed from labels, (**e**); 'APT5' was removed from labels. Corresponding data for Layers 4-6 is shown in Figure S8. Panels indicate 6 representative cell-types. Proteins significant in proteomics (FDR adjusted *p*-value < 0.05) are shown in red if their respective snRNA-seq FDR adjusted *p*-value < 0.3 and orange otherwise. Proteins not significant in proteomics are shown in grey. Source data are provided as a Source Data file.

seq (adjusted *p*-value < 0.0001), 42 out of 47 GO terms identified that were also significant in proteomics (adjusted *p*-value < 0.05) had the same direction of up/down-regulation. Major GO terms identified as down-regulated in proteomics were identified in snRNA-seq as well, such as mitochondria, oxidative phosphorylation and ATP

biosynthetic processes. However, several pathways diverged between proteomics and snRNA-seq. For example, kinases were strongly up-regulated in proteomics, whereas snRNA-seq showed a down-regulation of GO terms related to kinase binding and regulation of kinase activity in astrocytes and microglia and a minor up-regulation of

'NF-kappaB signal transduction' in VIP neurons. Whereas (small) GTPase binding/regulation was up-regulated in the proteomics data, but down-regulated in snRNA-seq data of microglial cells, astrocytes, and excitatory layer 5–6 intra-telencephalic neurons. RNA splicing/ processing pathways were down-regulated in proteomic, whereas snRNA-seq data showed up-regulation across inhibitory and excitatory neurons in snRNA-seq, but down-regulation in astrocytes. Ribosomes were down-regulated in proteomics data but snRNA-seq data showed a mixed pattern, with up-regulation in microglia and down-regulation in astrocytes and SST neurons.

We compared the effect sizes of proteins in selected pathways to their respective snRNA-seq effect sizes and found that genes with non-zero control vs schizophrenia differential expression exhibited generally similar trends across the proteomic and transcriptomic datasets (Fig. 4b–f, Figure S8). Because statistical power to detect differential expression between control and schizophrenia in the snRNA-seq dataset was generally lower, we applied an adjusted p-value ($\leq 0.3$) as used in the snRNA-seq study to achieve sufficient overlap with our proteomics dataset. We found meaningful trends-of-regulation that corroborate several pathways identified by our proteomics analysis. For example, ribosomal proteins were mostly down-regulated in the proteomics data and the corresponding genes measured by snRNA-seq in excitatory neurons and astrocytes were either down-regulated or unchanged. From $N = 10$ and $N = 16$ significant proteins from Layers 1–3 and 4–6 that overlapped with snRNA-seq data, respectively, 72% and 70% were found regulated in the same direction across all cell types in snRNA-seq data (considering all genes, including non-significant) (Fig. 4b). Similarly, there seems to be an equidirectional trend in both data modalities for the NADH complex with 58% ($N = 10$) and 68% ($N = 16$) co-regulation (Fig. 4c) for significant proteins from respective Layer sets, respiratory chain complex IV with 100% ($N = 4$) and 100% ($N = 3$) co-regulation (Fig. 4d) and proton-transporting ATP synthase complex with 86% ($N = 8$) and 77% ($N = 7$) co-regulation (Fig. 4e). Conversely, proteins related to RNA splicing only showed high co-regulation in astrocytes (Fig. 4f).

Importantly, comparative analysis of the transcriptome (Bast et al.[32]) and our proteomics analysis consistently highlights a specific downregulation of the ATP-generating machinery and supports a mitochondrial dysfunction phenotype in schizophrenia. Moreover, the proteomic analysis lends credit to the view that related highly ATP-dependent processes, such as those involved in synaptic transmission, and kinase signaling are also affected. Together, the proteomic and transcriptomic data provide a unique, complementary perspective by revealing both proteins and pathways that are co-regulated at multiple levels or that may be selectively affected post-translationally.

## Cell type expression patterns across enriched GO terms

To investigate whether schizophrenia-related GO terms from proteomic analyses (Fig. 3) exhibit cell type specificity we retrieved the relative gene expression levels for the respective proteomic GO term constituents from the Bast et al. snRNA-seq dataset[32]. A heatmap visualization of cell type-specificity scores for 34 BP terms significant in proteomics data revealed heterogeneity among cell types (Figure S9a). For individual genes within GO terms, we observed different patterns of cell type-enrichment. Among oxidoreductase activity-related genes, some (e.g., *GFOD1*, *KCNAB2*) are highly enriched in excitatory neurons whereas others (e.g., *ALDH4A1*, *ALDH6A1*, *GPD2*, *IDH2*) are astrocyte-enriched (Figure S9b). GTPase activator genes are partly cell type specific (e.g,. *SRGAP3*, *GIT1*, *RIN1*) while others exhibited expression across multiple cell types such as *DNM1L* in nearly any cell type, *PGAM5* in neurons, and *AGAP2* in excitatory neurons (Figure S9c).

## Comparative analysis with existing schizophrenia omics data

We compared the differentially expressed proteins from our proteomics datasets (Fig. 1b, c) with those reported in recent proteomic and genomic studies of schizophrenia and observed only limited overlap among schizophrenia-associated genes identified across studies (Fig. 5a, b).

MacDonald and co-workers applied a targeted proteomics approach using selected reaction monitoring (SRM) to quantify a priori selected proteins in both tissue homogenate and synaptosome preparations from the superior temporal gyrus (48 SCZ cases and 48 controls)[20]. They quantified 402 proteins in tissue homogenates and 155 proteins in synaptosomes of which 55 and 64, respectively, were differentially expressed at adjusted p-value < 0.05. Among the significant hits overlapping with our study, 14 out of 16 proteins were regulated in the same direction (up or down in schizophrenia). At the pathway-level, statistical power for GO analyses was reduced in their dataset due to the relatively low number of quantified proteins. Nonetheless, a subset of the pathways identified in our data (Fig. 3, Supplementary Data 5) were also observed by MacDonald et al. albeit with a lower coverage of respective proteins due to the difference in dataset depth; mitochondria, kinases, transmembrane ion transport, protein transport and cytoskeleton. Interestingly, their data revealed both homogenate-specific and synaptosome-specific proteomic alterations which demonstrated the unique SCZ signatures from both the global proteome and from targeted analyses of synaptic pathways, which argues for investigating both.

Aryal and co-workers applied (TMT) labelled proteomics for untargeted measurement of 8996 proteins in synaptosome preparations of the dorsolateral-PFC (35 SCZ cases and controls) and found 125 unique proteins differentially expressed at adjusted p-value < 0.05 (and 368 at threshold 0.1)[17]. Analysing the overlap between their data and this study using the same adjusted p-value cutoffs for consistency, we found 10 overlapping significant hits, of which 9 were regulated in the same direction (Fig. 5a, b). When considering proteins identified in both studies at adjusted p-value < 0.1 (reduced stringency to increase overlap), schizophrenia case-control effect sizes were strongly correlated; independent comparison between our results from Layers 1-3 and Layers 4-6 versus results from Aryal et al. showed 18 and 21 proteins exhibited a 0.89 and 0.79 Pearson correlation, respectively (Fig. 5c, Figure S10). Thus, the subset of proteins with at least moderate SCZ-related regulation in both studies showed similar up/down-regulation. However, when considering all proteins significant (adjusted p-value 0.05) in either study these correlations remain positive but decrease to 0.49 and 0.51 for Layer sets 1–3 and 4–5, respectively. The lower correlation for proteins beyond the top co-regulated proteins likely reflects the distinct tissue used in each study; we measured tissue homogenates whereas Aryal et al. used synaptosomes. Our study has a larger sample size (96 control and schizophrenia cases in total vs 70 for Aryal et al.) and our study measured 2 layer-sets per individual.

Additionally, there is agreement between the schizophrenia-associated GO terms identified in our dataset and the GSEA results by Aryal et al. Major classes of down-regulated proteins identified in our study, including mitochondrial inner membrane, oxidative phosphorylation, transporter complexes and ribosomes, were also identified as significantly down-regulated terms by Aryal et al. as well. Interestingly, while mitochondrial outer membrane was not significant in our dataset it was identified as significant by Aryal et al. albeit at much lower p-value (adjusted p-value 1.2e-8 vs 0.02). Similarly, the major classes of proteins up-regulated we observed, including GTPase binding, protein kinase activity (with related terms in the other study; kinase binding and regulation of protein kinase activity), (regulation of) autophagy, regulation of intracellular signal transduction and cytoskeleton organization, were also found to be up-regulated by Aryal et al. In contrast, Aryal et al. showed that 'phosphatase activity' is significantly up-regulated in the SCZ synaptosome, while we did not observe significant enrichment for this GO term in our dataset. Vesicle tethering, a main result in synaptosomes reported by Aryal et al. was

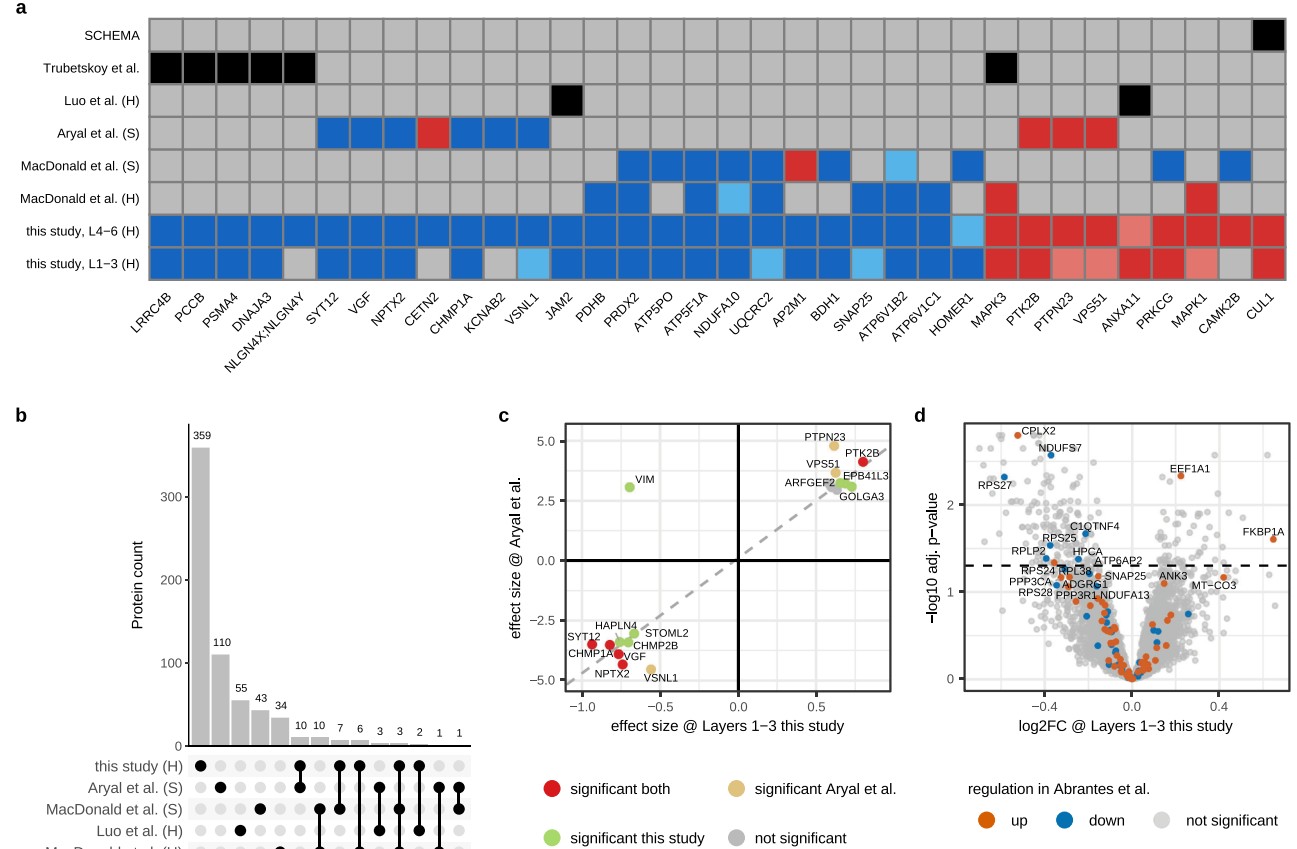

**Fig. 5 | Comparative analysis of data in this study with previously published schizophrenia (SCZ) datasets. a** Overview of significant hits from either Layer set in this study (at FDR adjusted p-value < 0.05) presented in Fig. 1b, c that overlap with any related SCZ proteomics (MacDonald et al.[20], Aryal et al.[17], Luo et al.[16]) or genetics (Trubetskoy[12], SCHEMA[14]) study. Up/down-regulated proteins are depicted in red/blue, with dark/light colors indicating significance at FDR adjusted p-values < 0.05 and < 0.1, respectively. Overlap with significant proteins reported in GWAS and pQTL studies is shown in black. Grey indicates no overlap. Where applicable, homogenate (H) and synaptosome (S) biochemical preparations are indicated. **b** Overlap of significant proteins (FDR adjusted p-value < 0.05) between proteomics studies is sparse. **c** A high degree of similarity in dysregulation is observed for 48 proteins differentially expressed at FDR adjusted p-value < 0.1 in both our study and Aryal et al. The x-axis shows SCZ effect sizes from Layers 1-3

(Fig. 1b). The y-axis shows effect sizes reported by Aryal et al. FDR adjusted p-value < 0.05 was used to classify proteins as significant; red: significant in both studies, green: significant in either study, grey: not significant in either. The dashed line shows a trendline fitted by robust linear regression. **d** Control vs SCZ analysis of Layers 1-3 presented in Fig. 1b, with proteins color-coded by their regulation after Olanzapine administration in mice as observed with RNA sequencing by Abrantes et al.[36]. Up/down-regulated proteins (FDR adjusted p-value < 0.05 in Abrantes et al.) are depicted in orange/blue. Note the anti-correlation of some proteins (e.g., synaptic proteins *CPLX2*, *SNAP25*) observed with lower abundance in SCZ patients in this study (x-axis) and the opposite effect observed for Olanzapine treatment in mice by Abrantes et al. (color). Corresponding data for Layers 4-6 is shown in Figure S10. Source data are provided as a Source Data file.

not significant in our homogenate data either although the direction of regulation was the same.

Interestingly, several observations emerged from our GO analyses that were not detected in the studies by MacDonald and Aryal. The GO term protein kinase activity has a strong effect in our dataset (nearly all proteins are up-regulated, Fig. 3c) but this ontology term showed no regulation in the synaptosome data from Aryal et al. (GOMF_PRO-TEIN_KINASE_ACTIVITY p.adj 0.77, their Table S3), was not observed in homogenate data from MacDonald et al. and showed weak regulation in synaptosome data from MacDonald et al. (p.adj 0.29, their eTable 10). Likewise, the GO terms RNA processing and splicing, that show a clear regulation in our study with most proteins having a lower abundance in SCZ (Fig. 3c). were neither enriched in Aryal et al. (p.adj 0.999), nor reported in either of the MacDonald datasets.

Taken together, we extend the prior characterizations of SCZ-dysregulated proteomes by MacDonald and Aryal by applying untargeted data-independent acquisition (DIA) proteomics of 5243 unique proteins in DPFC tissue homogenates (47 SCZ cases and 49 controls). This approach expands the scope of measurement in similar tissue as performed by MacDonald et al. and provides

orthogonal insights to the work by Aryal et al. that used synaptosome preparations. Despite the sparse overlap at the individual protein level among these three studies, the significant hits we identified are largely regulated in the same direction (Fig. 5). When combined with the observed pathway-level concordance across datasets, this suggests that these studies are complementary and collectively generate biologically meaningful leads on SCZ dysregulation. Our dataset contributes many significant proteins not detected in previous studies (Fig. 5b) and highlights strong regulation across a wide range of pathways (Figs. 3, 6).

Luo et al.[16] integrated proteomics and genomics data from 45 frontal cortices of schizophrenia cases[16]. Comparison of our control vs schizophrenia statistics with their reported top 60 priority genes, obtained from integration of five proteomic and genomic data sets, showed minor overlap (Fig. 5a, b). Notable shared proteins between both studies include *JAM2*, a cell-cell adhesion protein that inhibits myelination in the somatodendritic compartment[34] and *ANXA11* a phospholipid binding protein annotated in GO against 'vesicle-mediated transport' and 'response to oxygen-containing compound' that can bind S100 proteins[35].

We also compared our schizophrenia differential expression proteomics data (Fig. 1b, c) with schizophrenia-associated genes reported by Trubetskoy et al. in large scale GWAS[12] and found 6 significant proteins from this study that overlapped with GWAS results (Fig. 5a). Building on the S-LDSC analyses that confirmed concordance between proteomic and genetic load in schizophrenia (Fig. 2c), we here highlight significant proteins that intersect both datasets. Our GO analyses identified many protein kinases up-regulated in schizophrenia (Fig. 3c) including *MAPK3*, a component of the MAP kinase signal transduction pathway and a GWAS supported hit. Additionally, we identified 11 proteasome subunits as significantly down-regulated (Fig. 3e) among which *PSMA4* overlapped with GWAS results. Other major protein classes identified in our GO analyses that are confirmed by GWAS hits include mitochondrial protein *PCCB* that is involved with ATP binding, trans-synaptic adhesion protein *LRRC4B* that regulates the formation of synapses, and *DNAJA3* which is localized to mitochondria and postsynapses and is involved in GTPase regulation and tyrosine kinase binding.

From the 32 genes (at 5% FDR) with SCZ-associated rare coding variants that were previously reported by the SCHEMA study[14], *CUL1* was the only significant protein in our proteomics data (Fig. 5a). This up-regulated protein is part of E3 ubiquitin-protein ligase complexes together with *SKP1* (significantly down-regulated at adjusted *p*-value 0.02 in both Layer sets) and *DDB1* (down-regulated at adjusted *p*-value 0.2 in both Layer sets), is involved in the regulation of the cell cycle, and functions upstream of proteasomes (which are down-regulated in our data). Its paralog, *CUL3*, is also up-regulated in our data (adjusted *p*-value 0.03 and 0.02 in Layers 1–3 and 4–6, respectively). Beyond *CUL1*, another noteworthy intersection of SCHEMA and our proteomics data is the (non-significant) up-regulation of the kinase *TRIO* (adjusted p-value 0.21 and 0.06 in Layers 1–3 and 4–6, respectively), which falls within the general GO classification of kinases that we found up-regulated in SCZ (Fig. 3).

### Antipsychotic treatment effects on differentially abundant proteins

To investigate whether proteins dysregulated in our study are potential targets of drug treatment we compared our proteomic data to Abrantes et al.[36]. In their study, Abrantes and colleagues examined the effect of chronic exposure to olanzapine for its effect on single-cell gene expression in the mouse brain. Several genes found as treatment-affected in their dataset, were regulated in the opposite direction compared to our study of the schizophrenia human brain (Fig. 5d, Figure S10), suggesting potential treatment targets. Notably, complexin-2, which is down in human brain is up-regulated after olanzapine in mice. Similarly, the human schizophrenia down-regulated calcineurin protein phosphatase (*PPP3CA*) was up-regulated by olanzapine treatment in mice[36] and is known to be affected by various antipsychotics[37]. Also, expression of some calcium-regulated kinases found in our study (e.g., *CAMK2A*), is also influenced by antipsychotic treatment[38].

### Discussion

We performed a case-control proteomics study on 192 post-mortem tissue sections of the dorsal prefrontal cortex from 96 individuals, including 47 clinically diagnosed schizophrenia cases and 49 healthy individuals. While many proteins were differentially expressed between the upper and deeper cortical Layers (Fig. 2a), we found a very strong correlation of the protein effect sizes between control and schizophrenia samples when tested independently per cortical layer (Fig. 1d). Across these independently-generated cortical layer datasets, we identified a total of 387 proteins differentially expressed between controls and schizophrenia. This significantly regulated set of proteins is enriched for common genetic variants for schizophrenia, enriched in GWAS-enriched GO terms, and contains several proteins identified by

pQTL analysis. GOAT gene ontology analysis provided evidence for group-wise regulation of several major protein categories, emphasizing downregulation of mitochondrial oxidative respiration and the proteasome, upregulation of kinases vs up- and down-regulation of phosphatases, and upregulation of (small) GTPases. SynGO analysis further supports the notion of synaptic dysfunction in schizophrenia, with major regulators of pre- and postsynaptic function compromised, however with limited overlap of the specific proteins identified here and previous genetic data.

In contrast to previous studies, we analyzed differential expression of two independently measured sets of cortical Layers (1–3 and 4–6) of the DPFC within each patient and control. This revealed an exceptionally high correlation between the two layer groups and showed an absence of differential disease-related expression, i.e., proteins that exhibit a strong SCZ-related change in abundance levels in one cortical layer set exhibited a similar effect size in the other layer set.

Rigorous quality control revealed that well-known potential confounders, such as post-mortem delay time, age, sex, and various technical issues, did not bias our results. Also, we did not observe significant differences attributable to clinically relevant variables, including alcohol, nicotine, substance use, and medication in the subsets of cases tested. Notably, we identified upregulation of the nicotinic acetylcholine receptor nicotinic alpha 4 subunit (*CHRNA4*), suggestive of the impact of nicotine exposure through smoking. A singular source of variation was attributed to the source (lab/site) where the tissue originated, for which we employed linear modeling to correct for the between-site differences.

Traditional gene ontology overrepresentation analyses rely on the significance of individual proteins and the chosen p-value cutoff, often overlooking trends in the regulated expression of larger protein groups. This limitation is particularly relevant in schizophrenia, where many underlying genes contribute with a small effect size. To address this, we applied the parameter-free GOAT algorithm to identify GO terms significantly enriched for proteins with a strong effect size. The GOAT enrichment score for a given GO term utilizes a compound score based on all gene constituents (not only significant hits), assigning greater weight to strong effect sizes and thereby detects enrichment trends where many genes within a pathway show collective enrichment in either up- or down-regulation.

The analysis of the top list of down-regulated proteins revealed a subset of key enzymes in the complexes I, II, IV, and V of the mitochondrial electron transport chain, specifically affecting ATP production. For instance, cytochrome c oxidase 5 A, and -B (*COX5A*, *COX5B*), facilitate the transfer of electrons from cytochrome c to molecular oxygen and contribute significantly to the proton electrochemical gradient across the inner mitochondrial membrane, a process essential for driving ATP synthesis. Adaptations in energy production genes, including *COX5A/B*, are thought to have played a role in the emergence of the energetically expensive human neocortex[39], and even minor dysfunctions could therefore have a notable impact on cognitive function. Also, *SDHA* and *SDHB* part of complex II, facilitate the integration of two significant pathways within mitochondria, namely, the oxidation of succinate to fumarate as a vital constituent of the TCA cycle and the conversion of ubiquinone to ubiquinol. Both processes are indispensable for oxidative phosphorylation and ATP generation[40].

A role for oxidative phosphorylation in the pathogenesis of schizophrenia was first proposed by Takahashi (1953), who observed a reduced oxygen uptake in brain tissue from schizophrenia patients[41]. Subsequent clinical studies utilizing positron emission tomography (PET) unveiled a decreased oxidative metabolism in the frontal cortex of schizophrenia patients[42,43]. Transcriptomic analysis of 104 post-mortem prefrontal cortex samples showed that genes related to metabolism and oxidative stress differentiated almost 90% of schizophrenia patients from controls[15]. More recently Cavelier et al. showed

reduced activity of mitochondrial complex I, specifically in the prefrontal cortex[44], with a reduction in COX activity of 43%. Notably, this decrease in COX activity did not correlate with changes in the total amount of mitochondrial DNA, suggesting that the alterations were not due to differences in the number of mitochondria[44]. This aligns with our observation that predominantly mitochondrial inner membrane proteins' abundance is changed in schizophrenia, while those of the outer membrane are unchanged, an expected scenario if changes are not only related to differences in the number of mitochondria. Consistent with the finding of Ghosal et al.[45], who reported lower levels of ATP in the dorsolateral prefrontal cortex of individuals with schizophrenia, we observed downregulation of all ATP5 subunits of the mitochondrial ATP synthase. Of interest is also the lower expression of gamma synuclein (*SNCG*) as lower levels have been shown to negatively impact ATP synthase efficiency[24]. While these alterations in mitochondrial function could potentially lead to reduced ATP production, it remains unclear whether this is a primary driver or a secondary consequence of the disorder. Either independent, or as a result from decreased ATP production there is a marked down regulation of ribosome constituents.

Using GOAT, we established the groupwise dysregulation of kinases, small GTPases, and proteins of the splicing machinery. Notably, the observed kinase upregulation co-occurs with the downregulation of several phosphatases, potentially synergistically enhancing phosphorylation. Regarding kinases, it was previously found that dysregulation of the AKT pathway may contribute to abnormalities in the development and functioning of GABAergic interneurons, which are potentially key cellular components of neural circuits disrupted in schizophrenia[46,47]. Post-mortem studies of individuals with schizophrenia have revealed reduced expression of genes involved in GABA synthesis and transport, as well as decreased density and altered morphology of GABAergic interneurons in various brain regions[48–50]. Additionally, dysregulation of the MAPK pathway has been implicated in schizophrenia[51], with evidence suggesting that alterations in MAPK activity may contribute to abnormal neurodevelopment and cognitive deficits in the disorder. We found *MAPK3* and *MAPK1* to be the most highly up-regulated kinases.

Genetic studies also support a role of kinase dysregulation in schizophrenia. For instance, GWAS studies have identified gene variants encoding kinases that are associated with an increased risk of developing schizophrenia, including the genes encoding protein kinase C alpha (*PRKCA*)[52] and encoding protein kinase C epsilon (*PRKCE*)[53]. In our dataset, protein kinase C family members *PRKCE* and *PRKCG* were significantly associated with schizophrenia. Moreover, other studies have identified variations in genes encoding additional kinases, such as *CAMK2A*, *PRKCD*, *PRKAG2*, and *PAK3*, which are also associated with an increased risk of developing schizophrenia[54–56]. Together, these genetic studies suggest that variations in genes encoding kinases may contribute to the risk of developing schizophrenia, and that dysregulation of kinase activity may play a role in the pathophysiology of the disorder.

SynGO analysis revealed that 32% of significant proteins are known pre- and/or postsynaptic proteins. Among the down-regulated proteins, both presynaptic complexin-1 and complexin-2 were identified. Complexins are key regulators of the activity of soluble N-ethylmaleimide-sensitive factor attachment protein receptor (SNARE) complexes, which modulate synaptic vesicle exocytosis. They can exert both inhibitory and stimulatory effects on exocytosis by clamping trans-SNARE complexes in a prefusion state and promoting conformational changes to promote membrane fusion. It is noteworthy that complexin-1 is predominantly expressed in inhibitory neurons, whereas complexin-2 is mainly expressed in excitatory neurons[57]. Expression differences of complexins have been previously observed in schizophrenia[57,58], and in schizophrenia patients, these differences may vary between brain regions[57]. Additionally, single-nucleotide

polymorphisms distributed over the complexin-2 gene were found strongly associated with current cognitive performance of schizophrenic individuals[59].

When compared with the mouse study by Abrantes et al.[36], which examined the effects of chronic exposure to olanzapine on gene expression in the mouse brain, we found that several genes regulated in their study were altered by treatment in the opposite direction. These genes/proteins may be relevant targets for drug intervention. Notably, complexin 2, which we find down-regulated in the human schizophrenia brain, is up-regulated after olanzapine treatment in mice. If Olanzapine were to act on the same molecular targets in the human brain, this would suggest that upregulation of the presynaptic release machinery might be relevant in schizophrenia pathology and/or treatment of patients. Similarly, the calcineurin protein phosphatase (*PPP3CA*), which is down-regulated in schizophrenia is affected by olanzapine treatment in mice[36], and by various antipsychotics[37]. Furthermore, some calcium-regulated kinases, such as *CAMK2A*, which were found to be affected in our studies, also exhibit altered gene expression in response to antipsychotic treatment[38].

Furthermore, a substantial group of postsynaptic density proteins were found to be enriched in schizophrenia, typically including *CAMK2A*, *HOMER1*, *DLGAP1*, and *LLRC4B*. *DLGAP1* was previously identified based on SNPs associated with schizophrenia[60], while *LLRC4B* was recently identified by GWAS to be associated with schizophrenia[12]. Each of these proteins is known to have a significant impact on synapse function (Figure S11).

When examining subcellular distribution of differentially regulated proteins in schizophrenia, the synapse is not the primary subcellular compartment exhibiting differential expression (Fig. 3, Figure S11). Instead, processes such as clathrin-mediated endocytosis, endosomal trafficking, cytoskeleton organization and associated motor proteins, protein synthesis and breakdown, and a large group of nuclear proteins are affected. A striking observation is the downregulation of clathrin light chain proteins A and B (*CLTA* and *CLTB*), in combination with the upregulation of clathrin heavy chain (*CLTC*), dynamins (*DNM1/2/3*) and dynamin-like 1 (*DNML1*) in schizophrenia donors. These proteins are key components of the clathrin-mediated endocytosis and clathrin-dependent membrane and protein trafficking, processes previously postulated to have a role in schizophrenia[61]. Related to this, proteasomal degradation and autophagy are also affected, with many proteasome subunits found to be down-regulated. In post-mortem schizophrenia tissue, dysregulation of ubiquitin–proteasome-related genes has been observed[62]. Similarly, genetic studies found evidence for an association of the ubiquitin–proteasome pathway with psychotic disorders, as well as correlations of several ubiquitin–proteasome pathway proteins with a clinical dimension in schizophrenia[38,63], which is consistent with our finding that many proteins of the ubiquitin system are dysregulated clathrin-mediated processes also depend on vesicular transport. Our data shows the downregulation of actin-bound proteins, e.g., motor proteins involving myosin light chains (*MYL6/6B*, *MYL12B*, *MYL9*), and proteins related to the organization of the actin cytoskeleton, such as capping proteins (*CAPZA1* and −2, *CAPZB*), actin organizers, such as the ARPs (*ARPC5*, *ARPC5L*), WIPF2, tropomyosins 1-4, tropomodulin 1. Notably, the most prominently up-regulated protein in our study is the microtubule alpha subunit *TUBA1B*. For microtubule associated proteins we find a significant upregulation for axonal kinesin motor proteins (*KIF1A*, *KIF2A*). Among nuclear proteins, we find alterations in RNA-binding proteins which are involved in splicing, such as *HNRNPC*, *TRA2B*, *RALY*, *SRSF3* (all down-regulated), or in nucleo-cytoplasmic transport including *RAN* (up-regulated).

In comparison to the snRNA-seq data of the companion study by Bast et al., which used consecutive DPFC tissue sections, we found agreement regarding the downregulation of mitochondrial proteins, ribosomes, NADH dehydrogenase complex assembly and proton-

transporting ATP synthase complex (Fig. 4). These findings suggest that changes in metabolic proteins may arise from changed gene expression. In contrast, specific protein groups appear uniquely differentially regulated at the proteome level, including the strongly up-regulated kinases and the dysregulation of synapse-specific proteins, such as the complexins. Our data indicate a potential feedback mechanism between the downregulation of mitochondrial ATP production and the broad upregulation of the kinases, which use ATP as substrate pointing to a likely role for translational or posttranslational control. Overall correlation between transcriptome and proteome data on the individual gene/protein level is rather low.

Comparison of our proteome data with recent GWAS data (Fig. 5a) revealed limited overlap, which also holds for the 15 synaptic 'core group' proteins previously defined in the most recent GWAS[12]. Although this may reflect our much smaller cohort size, interestingly, our proteomics study is carried by a GWAS signal as evidenced by LDSC. For the rare variants of the exon sequencing in the SCHEMA study[14] only *CUL1* is found in overlap. Taken together, sequence variants with high penetration are infrequent in the population and may therefore not be captured in the current study, while common GWAS variants may not yield the measurable change needed to be detected at the level of gene and protein expression.

It is noteworthy that protein groups identified in this study were partially confirmed by comparing to recent data on the synapse proteome in schizophrenia cases. We found high agreement (Pearson's rho 0.89 and 0.79 for each Layer set, respectively) with Aryal et al.[17] with respect to the set of proteins (18 and 21 for each Layer set, respectively) identified at adjusted *p*-value < 0.1 in our study and their synaptosome preparations of the dorsolateral prefrontal cortex (DLPFC) in schizophrenia patients. Similarly, 14 out of 16 significant hits that overlapped between our data and SCZ proteome analyses of homogenate and synaptosome preparations from the superior temporal gyrus reported by MacDonald et al.[20] exhibited equidirectional regulation. GO analyses from both studies identified down-regulation in schizophrenia cases for ontology terms related to mitochondria, transport and ribosomes and as well as up-regulation for GTPase binding, protein kinase activity, autophagy, signal transduction and cytoskeleton organization.

Although our study includes the largest set of schizophrenia patient brain tissue to date it is still small compared to recent GWAS genetics studies. In addition, it would be of specific interest if larger cohort studies would have the power to improve dissecting the potential effects of medication. Also, we studied the dorsal prefrontal cortex and it would be beneficial to include other brain regions in future studies, f.i. including those that provide modulatory input to the cortex.

In summary, this study provides the most comprehensive proteomics analysis of human schizophrenia brain tissue homogenates to date. We identified numerous significantly dysregulated individual proteins including entire functional groups such as kinases, small GTPases as well as key regulators of organellar function, including mitochondria, synapses, and the cytoskeleton. Our findings reveal protein dysregulation that is carried by a genetic signal highlighting several candidates with potential as targets for treatment. Our study provides a robust foundation for further exploration into the molecular mechanisms of schizophrenia. Amongst others, the consistent mitochondrial dysfunction observed in the prefrontal cortex emerges as a compelling focus for future research and therapeutic intervention.

## Methods
### Ethics declaration and cohort information
Post-mortem brain tissue of the dorsal prefrontal cortex (DPFC) of 47 clinically assessed schizophrenia cases and 49 healthy controls were obtained from three sites. Cases and controls were selected based on clinical and neuropathological reports (evaluated by PFS). We obtained 3 controls and 8 cases from The Netherlands Brain Bank (NBB, denoted as BA in sample metadata). Provision of samples and study design was approved by the NBB's Tissue Advisory Board. We obtained 19 controls and 13 cases from the brain sample collection of Dr Andrew Dwork (denoted as AA in sample metadata). Provision of samples from the Macedonian/New York State Psychiatric Institute Brain Collection and study design was approved by the New York State Psychiatric Institute Institutional Review Board. We obtained 27 controls and 29 cases from the brain sample collection of Craig Stockmeier (denoted as CB in sample metadata). Provision of samples and study design was approved by the IRB of the University of Mississippi Medical Center (Protocol 1999-1002) and the University Hospitals Case Medical Center (Protocol 11-88-233). All brain tissue was collected from donors with written informed consent for brain autopsy and the use of brain tissue and clinical information for research purposes, complying with all relevant regulations at each site of sample collection.

Controls and schizophrenia donors used for proteomics had very little or no signs of neurodegeneration, including the absence of detectible amyloid β, tau protein, granulovacuolar degeneration, α-synuclein, or p62 (sequestosome1) pathology in the cortex. Controls were reported to be cognitively healthy. We aimed to collect a balanced number of male and female donors for control (22 female, 27 male) and schizophrenia (24 female, 23 male) cases to avoid sample collection bias. Sample numbers were too low to confidently establish differential SCZ regulation between sexes. Clinical data, including post-mortem delay time (PMD) of autopsy, age, and sex for all samples is provided (Supplementary Data 1). We also documented medication and other drug use of all brain donors. As part of quality control procedures, schizophrenia cases were also checked for Alzheimer's Disease pathology that might be present as amyloid β deposits, neurofibrillary tangles and neuritic plaques conform the ABC score[21–23,64].

### Tissue sectioning for anatomical reference from the dorsal prefrontal cortex
From each donor, 4 sections (10 μm) were used for pathological and anatomical reference stain for laser-capture microdissection (LCM) to determine Layers 1-3 and 4-6. Sections were mounted on SuperFrost slides, air-dried, and fixed in 100% acetone for 5 minutes. After air-drying, the tissue was wetted with sterile phosphate-buffered saline (PBS), pH 7.4. Next, endogenous peroxidase was blocked in 0.3% $H_2O_2$ in PBS for 30 mins. Sections were incubated with IC16 (amyloid beta - a kind gift of Prof Dr Korth, Heinrich Heine University, Düsseldorf, Germany) 1:800, AT8: tau (Pierce Biotechnology, Rockford, IL); 1:800, p62 (clone 3/P62 LCK LIGAND, BD Transduction Laboratories, San Jose, CA, USA) 1:1000, and SMI-32 (Biolegend, San Diego, CA, USA) 1:8000 for an hour at room temperature. Sections were washed in PBS and incubated with secondary (Envision, DAKO) for 30 mins, and after washing incubated in DAB (DAKO), cover-slipped with Quick D (Duiven, The Netherlands). Each staining included appropriate negative and positive controls.

Local age-related protein aggregation in the DPFC were assessed in the IC16, AT8 and P62 stainings. When local pathology was observed, the tissue was excluded. The SMI-32 staining was used as an additional reference for Layer specificity for LCM dissection, where Layers 1-3 and Layers 5-6 were isolated separately using LCM.

### Tissue sections for LCM
Sections (10 μm) of fresh-frozen human DPFC were mounted on PEN-membrane slides (Leica Wetzlar, Germany), air-dried and fixed in 100% acetone for 5 min. After air-drying the tissue was wetted with sterile phosphate-buffered saline (PBS), pH 7.4. Sections were thoroughly washed in ultra-pure $H_2O$ and incubated with 1% (w/v) toluidine blue (Fluka Analytical, Buchs, Switzerland) in ultrapure $H_2O$ for 5 min as a counterstain. Sections were then washed in ultra-pure $H_2O$ twice for 1 min and twice in 100% ethanol for 1 min and air dried. The Toluidine

blue staining usually revealed the Layer 4 granular layer morphology, and SMI-32 revealed the larger Layer 3 neurons. If the Toluidine blue staining on the slides themselves did not reveal the exact border, the SMI-32 staining was used in parallel to determine the dissected area. The lower border of Layer 6 was determined by Toludine blue staining (Figure S2).

## Protein in-gel digestion

Micro-dissected tissue lysates were denatured and reduced by incubation at 95 °C for 5 min, followed by alkylation of free sulfhydryl groups by incubation with 50 mM iodoacetamide for 30 min at room temperature. Samples were loaded onto a 10% acrylamide gel and resolved for approximately 1 cm.

The gels were fixed overnight and stained with colloidal Coomassie Blue G-250. Sample lanes were cut into 1 mm$^3$ cubes, transferred to a MultiScreen HV 96-well filter plate (Millipore) and destained until clear using repeated applications of 50 mm NH3HCO3 in 50% acetonitrile. After dehydration with 100% acetonitrile, each well was supplemented with 0.67 ug MS grade Trypsin/Lys-C (Promega) in 50 mM NH4HCO3 and tryptic digestion was performed overnight at 37 °C within a humidified incubator. Tryptic peptides were extracted and pooled by two incubations with 0.1% trifluoroacetic acid in 50% acetonitrile, dried by SpeedVac and stored at −80 °C.

## Liquid chromatography–mass spectrometry (LC-MS) Analysis

Each sample of tryptic digest was redissolved in 0.1% formic acid and the peptide concentration was determined by tryptophan-fluorescence assay[65]; and 50 ng of peptide was loaded onto an Evotip (Evosep). Peptide samples were separated by standardized 30 samples per day method on the Evosep One liquid chromatography system, using a 15 cm × 150 μm reverse-phase column packed with 1.9 μm C18-beads (EV1106 from Evosep) connected to a 20 μm ID ZDV emitter (Bruker Daltonics).

Peptides were electro-sprayed into the timsTOF Pro 2 mass spectrometer (Bruker Daltonics) equipped with CaptiveSpray source and measured with the following settings: scan range 100–1700 m/z, ion mobility 0.6 to 1.6 Vs/cm$^2$, ramp time 100 ms, accumulation time 100 ms, and collision energy decreasing linearly with inverse ion mobility from 59 eV at 1.6 Vs/cm$^2$ to 20 eV at 0.6 Vs/cm$^2$.

Operating in data-independent acquisition (DIA) PASEF mode, each cycle took 1.8 s and consisted of 1 MS1 full scan and 16 DIA-PASEF scans. Each DIA-PASEF scan contained two isolation windows, in total covering 400-1201 m/z (1 Th window overlap) and ion mobility 0.6 to 1.43 Vs/cm$^2$.

## Proteomics Data Analysis

DIA-PASEF raw data were processed with DIA-NN 1.8[66]. An in-silico spectral library was generated from the UniProt human proteome (SwissProt and TrEMBL, canonical and additional isoforms, release 2021-04) using Trypsin/P digestion and at most 1 missed cleavage. Fixed modification was set to carbamidomethylation (C) and variable modifications were oxidation (M) and N-term M excision (at most 1 per peptide). Peptide length was set to 7-30, precursor charge range was set to 2-4, precursor m/z was limited to 380–1220, both MS1 and MS2 mass accuracy were set to 10 ppm, double-pass-mode and match-between-runs were enabled. Protein identifiers (isoforms) were used for protein inference. All other settings were left as default.

MS-DAP 1.0.4[67] was used for downstream analyses of the DIA-NN results. Filtering and normalization were applied to all sample groups in the dataset. Peptide-level filtering was configured to retain only peptides that were confidently identified in at least $N = 3$ and at least 50% of samples per sample group. Peptide abundance values were normalized using the VSN algorithm, followed by protein-level mode-between normalization. The MaxLFQ algorithm was used to create a protein-level data matrix, which was subsequently used as input for statistical analyses.

Limma[68] 3.52.2 was used to perform linear regression analyses, followed by application of the DEqMS[69] 1.14.0 algorithm to adjust the confidence levels of proteins according to their number of quantified peptides. Specifically, to test differential protein expression between controls and schizophrenia cases we specified source (/origin), gel ID, PMD, and age as covariates. To test differential expression between Layer sets, for the subset of individuals without SCZ, the same covariates were used. The within-block (2 Layer sets of the same individual) correlation was estimated at 0.71 by limma. The integration of these tools within the MS-DAP pipeline was used to perform the analyses. The resulting p-values were adjusted for multiple testing using the Benjamini−Hochberg False Discovery Rate (FDR) procedure.

## Gene ontology analysis

The GOAT[31] R package version 1.1.2 was used to perform gene set enrichment analyses with the GOAT algorithm. GO gene sets were obtained from the NCBI gene2go dataset (downloaded on 2025-01-01). GO enrichment in each Layer set was tested independently, using the respective effect size-derived gene scores to test for enriched gene sets that contained ≥10 and ≤1500 genes that overlapped with the input gene list. Multiple testing correction was independently applied per gene set 'source' (i.e., GO_CC, GO_MF, GO_BP) using Bonferroni adjustment and subsequently all $p$-values were adjusted (again) using Bonferroni adjustment to account for 3 separate tests across sources. The significance threshold for adjusted p-values was set to 0.05. The SynGO[33] analyses (Fig. 3f, Supplementary Data 6) were performed using the https://www.syngoportal.org webtools (release 1.2), with all significant hits across both Layer sets as foreground and all proteins from our proteomics dataset as the background.

## Protein quantitative trait loci (pQTL) analysis

Data from donors with both proteomic and SNP genotype data were analyzed. All samples were genotyped with Illumina GSA SNP arrays at SciLifeLab (Uppsala, Sweden). To eliminate the influence of genetic background, we limited pQTL analysis to donors of European ancestry (72). SNP quality control included: using standard thresholds in RICOPILI[70] (2019) to remove outlying individuals and SNPs and for imputation to Haplotype Reference Consortium. Basic filters included RICOPILI: INFO score > 0.1 and MAF > 0.005, genotype calls with P > 0.8, SNP missingness <0.02, removal of multi-allelic SNPs and indels, and retention of SNPs with allele count > 10. The final number of SNPs was 1,918,447.

pQTL analysis was performed using QTLtools[71] (v1.3.1) with nominal test for 5243 proteins in the protein abundance matrix (after combining the protein abundances in Layers 1-3 and Layers 4-6 for each donor). We tested all SNPs that passed QC within a cis-window of ±1 MB around the transcription start site (TSS) for the gene of the measured protein (total of 6,587,394 SNP-protein tested pairs across 4999 proteins). Covariates adjusted in each test included genotyping batch, sex, case-control status, and the first 3 genotyping principal components. To address multiple-comparisons, q-values were calculated for all tested pairs using the qvalue R package.

## Linkage Disequilibrium Score Regression analysis

Stratified linkage disequilibrium score regression (S-LDSC) was used to examine whether differentially expressed proteins are enriched for common genetic risk of psychiatric disorders and brain traits. We included the most recent European-ancestry GWAS for schizophrenia (SCZ)[12], bipolar I disorder (BiP)[27], major depressive disorder (MDD)[28], autism spectrum disorder (ASD)[29], and intelligence quotient (IQ)[30]. We defined three annotation groups based on the findings of the differentially expressed proteins: 1) the top 200 up-regulated proteins, 2) the top 200 down-regulated proteins, and 3) 200 selected proteins

equally distributed around zero effect size (i.e., no significant differences between cases and controls). SNPs within each annotation were compared to the other SNPs for the per-SNP heritability of the included traits in the stratified LD-score analysis with 53 baseline annotations[72]. The MHC region was excluded due to its confounding LD structure. FDR was used to correct for multiple comparison of the annotations for each trait.

### Cell type enrichment analysis

Cell type enrichment analysis can help to stratify data from mixed cell populations, without the need for physical cell sorting[73,74]. As single cell proteomic data sets are for now unattainable, we set out to identify cell type enrichment in our schizophrenia protein signatures based on single-nucleus RNA sequencing data from adult post-mortem DPFC tissue from a subset of 42 control and 41 schizophrenia samples of our cohort donors as used here for proteomics[32]. From this snRNA-seq dataset we obtained the expression level of each gene across 16 cell types. These values were subsequently rescaled linearly between 0 and 1 by dividing the abundance value of each gene in each cell type by its respective maximum value across all cell types. These data are shown in a heatmap for selected proteins in Figure S9b, c. The cell type enrichment score of a GO term was then computed as the 20% trimmed mean of cell type enrichment values of its respective genes (Figure S9a).

### Reporting summary

Further information on research design is available in the Nature Portfolio Reporting Summary linked to this article.

## Data availability

The mass spectrometry proteomics data have been deposited in the ProteomeXchange Consortium via the PRIDE partner repository[75] with the dataset identifier PXD058441. Source data are provided with this paper.

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

## Acknowledgements

This work was supported by grants from the Dutch Research Council (NWO) research programme sectorplannen Biologie (51499 to F.K.), the Swedish Research Council (Vetenskapsrådet) (D0886501 to P.F.S.), the IDeA COBRE award from the National Institute of General Medical Sciences (P30 GM103328 to C.A.S.), the European Research Council

advanced grant (ERC-2018-AdG GWAS2FUNC 834057 to W.P.L.). The authors thank the Netherlands Brain Bank (Amsterdam, the Netherlands) for supplying human brain tissue. The authors gratefully acknowledge contributions of the Cuyahoga County Medical Examiner's Office, Cleveland, OH, USA, and the families of the deceased.

## Author contributions

A.B.S., F.K., and P.F.S. designed the experiments. A.A.D., R.V.K., and Y.G. performed the experiments. F.K., W.P.L., S.Y., R.K., L.B., and P.F.S. performed data analysis. F.K. and A.B.S. interpreted the results. A.J.D. and C.A.S. provided samples, and A.A.D. performed the pathological characterization. A.B.S. and F.K. wrote the manuscript; P.F.S., M.V., and J.H.L. made intellectual contributions and contributed to the writing of the manuscript. All authors read and approved the final manuscript.

## Competing interests

P.F.S. reports the following potentially competing financial interest: Neumora Therapeutics (advisory committee, shareholder). To the best of his knowledge, these are unrelated to this paper/project. The remaining authors declare no competing interests.

## Additional information

[1]Department of Molecular and Cellular Neurobiology, Center for Neurogenomics and Cognitive Research (CNCR), Vrije Universiteit (VU) Amsterdam, Amsterdam, Netherlands. [2]Department of Pathology, Amsterdam University Medical Center (UMC), Amsterdam, Netherlands. [3]Swammerdam Institute for Life Sciences, University of Amsterdam, Amsterdam, Netherlands. [4]Department of Complex Trait Genetics, Center for Neurogenomics and Cognitive Research (CNCR), Vrije Universiteit (VU) Amsterdam, Amsterdam, Netherlands. [5]Department of Medical Epidemiology and Biostatistics, Karolinska Institutet, Stockholm, Sweden. [6]Department of Medical Biochemistry and Biophysics, Division of Molecular Neurobiology, Karolinska Institutet, Stockholm, Sweden. [7]Department of Functional Genomics, Center for Neurogenomics and Cognitive Research (CNCR), Vrije Universiteit (VU) Amsterdam, Amsterdam, Netherlands. [8]Department of Human Genetics, Amsterdam University Medical Center (UMC), Amsterdam, Netherlands. [9]Department of Pathology and Cell Biology, Columbia University, New York, NY, USA. [10]Department of Psychiatry and Human Behavior, University of Mississippi Medical Center, Jackson, MS, USA. [11]Department of Genetics, University of North Carolina, Chapel Hill, NC, USA. ✉e-mail: guus.smit@vu.nl

