## [Transparent Peer Review file · Nature Communications]

Human brain prefrontal cortex proteomics identifies compromised energy metabolism and neuronal function in Schizophrenia

Corresponding Author: Professor August Smit

Version 0:

Reviewer comments:

Reviewer #1

(Remarks to the Author)

This is a very well written research report on novel and highly relevant proteomics data, in conjunction with genetic association studies, of a comparatively large sample of schizophrenia vs. healthy post mortem brains. Methodologically the experiments are diligently performed and the results are highly coherent both within the study itself and with previous studies. A particular strength is the demonstration of a downregulation of mitochondrial oxidative phosphorylation with aligns well with imaging studies. This study does advance the field significantly.

I have only a few points the answer to which, I believe, will further strengthen the paper:

1. Could the authors perform a probabilistic permutation or bootstrap test on their proteomic data control vs. schizophrenia? This may further reduce the rather complex list of disease-specific regulated proteins and would further validate their findings
2. How big was the overlap in findings between the three different brain collections (Venn diagram)?
3. On a more conceptual level, there is an important analysis missing. Schizophrenia is a merely clinical diagnosis which relies on rather broad symptoms in the sense of a dysfunctional common final behavioral pathway such as psychosis or negative symptoms. The different courses of disease and symptomatic shaping already suggest different underlying biological causes. Any biological findings – such as those in the present paper - will therefore be correlates to either this final common path and/or the specific biological underpinnings of different subsets. However, pooling specific data on discrete biological subsets will dilute them below statistical significance. A statistical analysis that takes into account different subsets or clusters, should therefore be performed to characterize such subsets. For example, the authors claim commonalities between schizophrenia and bipolar disorder 1 (page 11; S-LDSC study) – could this be one subset (among others)?

Other points:

- “The MHC region was excluded due to its confounding LD structure” (page 8). Could the authors specify, please, what they mean by confounding LD structure? The MHC locus has shown highest reliability in GWAS studies and potentially links neuroimmunological causes (as one biological subset) to schizophrenia.
- Figure 6 is not helpful but only confusing and should not be shown.
- the Bast et al 2024 reference is listed several times but is not listed in the references - has this paper been deposited somewhere? If the authors consider data in this paper as essential for the present paper, those results should be accessible one way or another, else there is no need to reference them at all.
- the absence of page numbers in the manuscript is an unnecessary negligence
- Fig S4-S6, S9, S11-S12 – the protein abbreviation codes are too small and not readable, I therefore do not see any sense

of presenting this. Either show something readable or do not show it at all.

Reviewer #2

(Remarks to the Author)

Here the authors detected (quantified?) 36,176 peptides from 6,293 proteins in laminar DPFC dissections, obtained from a cohort consisting of 47 schizophrenia and 49 healthy control subjects. Overall, the study appears to have been well conducted, with a few significant and notable concerns: 1. The use of in-gel digestion is no longer standard in the field, having been abandoned almost a decade ago for more robust approaches including FASP and Strap. 2. Cortical laminae are distinguished by both cellular and molecular composition as well as the extent to which they display synaptic alterations in schizophrenia. Thus, the decision to "pool" the layers is not justified, even if similar trends between case and controls are observed in the different layers.

However, the largest issue with this manuscript is that it does not add new findings to the current body of work in the field. Multiple groups (most recently MacDonald 2020, PMID 31642882 and Arylal 2023, PMID: 37171958) have reported alterations in the levels of mitochondrial, synaptic, proteasome, and cytoskeletal proteins in both tissue homogenates AND synaptosome enrichments. The submitted manuscript investigates only tissue homogenates. The proteomic profiling in the latter of these two reports was significantly deeper than the submitted manuscript (711,402 unique peptides that mapped to 8,996 proteins).

Likewise multiple published reports of brain protein quantitative trait loci in the context of psychiatric disease (including schizophrenia) exist, all of which were generated from larger cohorts. Most notably:

1. Luo 2024 (PMID: 38724566) "11,608 proteins across 268. Our analysis revealed 788 cis-acting protein quantitative trait loci associated with the expression of 883 proteins at a genome-wide false discovery rate <5%". This study includes cases from schizophrenia subjects and significant integration of GWAS and proteomics data in the context of schizophrenia.
2. Wingo 2021 (PMID: 33846625) "we integrated depression genome-wide association study (GWAS) results (N = 500,199) with human brain proteomes (N = 376)"

Version 1:

Reviewer comments:

Reviewer #1

(Remarks to the Author)

The authors have addressed the main issues raised. Two minor issues remain at this point:

Even though negative, the attempt to identify subsets in the gathered data is important and deserves being mentioned in a paragraph in the main paper with mentioning of the exact statistical/ mathematical procedures, rebuttal Figure 1 can go into Supplement.

Figure 6: letters still cannot be distinguished - but I do see the point that the authors make. May I propose to publish Figure 6 as is, but to refer to an additional electronic Supplementary Figure not as part of the Supplementary material but as a small poster-like extra Supplementary Figure so that interested readers are able to decipher the protein abbreviations.

Reviewer #2

(Remarks to the Author)

The authors have addressed my technical concerns and have done an excellent job of describing their findings as well as distinguishing them from previous work in the field. However, it is still this reviewers opinion that the findings, cohort, and approaches are not sufficiently novel to merit publication in this journal.

REVIEWER #1

“This is a very well written research report on novel and highly relevant proteomics data, in conjunction with genetic association studies, of a comparatively large sample of schizophrenia vs. healthy post mortem brains. Methodologically the experiments are diligently performed and the results are highly coherent both within the study itself and with previous studies. A particular strength is the demonstration of a downregulation of mitochondrial oxidative phosphorylation with aligns well with imaging studies. This study does advance the field significantly.

I have only a few points the answer to which, I believe, will further strengthen the paper:

1. Could the authors perform a probabilistic permutation or bootstrap test on their proteomic data control vs. schizophrenia? This may further reduce the rather complex list of disease-specific regulated proteins and would further validate their findings

We have implemented a bootstrapping procedure as suggested. Results are shown in the new Supplemental Figure 4 and described in the result section on page 9, line 259;

“Bootstrapping procedures were used to estimate the empirical null distribution of protein effect sizes that would be obtained if sample labels (i.e. assigned phenotype) were randomly assigned (Fig. S4). We found that even the smallest effect sizes from significant hits in our CON vs SCZ analyses were outliers in the empirical null, confirming that the reported significant hits are highly unlikely to result from an overinterpretation of noisy data.

2. How big was the overlap in findings between the three different brain collections (Venn diagram)?

Thank you, this type of data presentation was indeed missing from the manuscript; we have now added an upset plot to Figure 5 (panel b) to comprehensively describe the overlap in significant proteins (at 5% FDR) between each dataset. We have also included an additional SCZ proteomics study (MacDonald et al.) in this comparative analysis. As discussed in the updated result section, in the paragraph “Comparative analysis with existing schizophrenia “omics” data” (page 14-17); the overlap in reported significant proteins/genes between the studies is quite low, however, there is considerably more agreement between studies at the pathway-level (i.e., enriched GO terms reported in each manuscript).

3. On a more conceptual level, there is an important analysis missing. Schizophrenia is a merely clinical diagnosis which relies on rather broad symptoms in the sense of a dysfunctional common final behavioral pathway such as psychosis or negative symptoms. The different courses of disease and symptomatic shaping already suggest different underlying biological causes.

Any biological findings – such as those in the present paper - will therefore be correlated to either this final common path and/or the specific biological underpinnings of different subsets. However, pooling specific data on discrete biological subsets will dilute them below statistical significance.

A statistical analysis that takes into account different subsets or clusters, should therefore be performed to characterize such subsets.

We agree that there is heterogeneity in SCZ phenotypic outcome and that diagnosis remains challenging. In line with previous SCZ proteomics studies of MacDonald et al. and Aryal et al., we tested proteins for significant differences between the two discrete groups of individuals labeled “control” and “schizophrenia”. This approach identified proteins with a difference in mean expression (across heterogeneous proteomes of 47 SCZ vs 49 control cases). Following your comment we explored a more in-depth stratification of protein regulation between subgroups of patients and/or controls but we were not able to identify such subtypes.

1) The median protein Coefficient of Variation (CoV) between replicates from each sample group (control or SCZ, per cortical Layer set) was 24.9~25.3% in controls and 23~23.6% in SCZ cases (Fig. S3c). A similar trend is seen for the subset of significant proteins; protein variation between SCZ cases is not higher than for controls. If there was a pronounced increase in proteome heterogeneity between SCZ cases as compared to controls we would have observed a significant increase in CoV among SCZ cases, but instead we found a similar CoV across all sample groups.

2) To follow up on the reviewer’s comment we tested if subsets of SCZ cases have differentially expressed proteins (as compared to controls), using unsupervised (unbiased) clustering to identify potential “SCZ subpopulations”, which can then subsequently be compared to controls. For this, we clustered Layers 1-3 samples from all cases and visualized the sample-to-sample similarity in a heatmap (Rebuttal Figure 1, below). We found no clear clusters of SCZ samples, and controls are dispersed with SCZ samples throughout. Because the unsupervised sample clustering is driven by expression patterns across all proteins this suggests that only a minor portion of proteins is differentially expressed in SCZ and with low magnitude of change, i.e. proteome alterations in SCZ are not strong enough to drive the clustering. This is what we expect given the small changes in levels of a minority of proteins. Ergo clustering is relatively weak (see dendrogram) and probably the result of a combination of biological and technical variation (as seen in the 23~25% CoV). The same trends were observed in Layers 4-6 (Rebuttal Figure 2, below).

In the absence of convincing clustering of SCZ samples into subtypes, there is no hypothesis driven approach to subdivide the SCZ cases and test them against other sample groups/clusters. In the bootstrapping analyses that you suggested in another comment we did observe that the control-vs-SCZ outcome of our statistical model is robust; the significant hits reported as our main result have effect sizes that trump the 23~25% CoV (new analysis, Fig. S4). Taken together, we conclude that we

would need more statistical power to follow up on the subtype issue, hopefully in future studies that yield higher N numbers.

For example, the authors claim commonalities between schizophrenia and bipolar disorder 1 (page 11; S-LDSC study) – could this be one subset (among others)?

Our stratified LDSC analyses in Figure 2c indeed showed that top-regulated proteins in our schizophrenia proteomics dataset are enriched in previously published GWAS data from Bipolar disorder 1 (in addition to SCZ and IQ). Additionally, the prior proteomics study from Aryal et al. on synaptosome dysregulations in SCZ and Bipolar disorder 1 showed differentially expressed proteins from both disorders are correlated (PMID: 37171958, their Figure 1E, Pearson's rho 0.70). However, we cannot put this idea (are subsets of our SCZ patients more similar to BiP than others?) to the test because we cannot meaningfully discriminate between SCZ subtypes in our data (see previous rebuttal point).

Rebuttal Figure 1. Unsupervised clustering of Layer 1-3 samples using the Euclidean sample*sample similarity matrix. Only 1 clear cluster is observed (top-left), which is not specific to (/overrepresented for) any of the annotated metadata; diagnosis, nicotine dependence, age, cohort, sex, age, gel (i.e. potential technical batch). As reported, considering the annotated sample metadata we only observe apparent clustering related to 'origin' (this factor we took along in the modeling of the data for differential expression analysis).

Rebuttal Figure 2. Analogous to Rebuttal Figure 1, but here for samples from Layer 4-6.

Other points:

- “The MHC region was excluded due to its confounding LD structure” (page 8). Could the authors specify, please, what they mean by confounding LD structure? The MHC locus has shown highest reliability in GWAS studies and potentially links neuroimmunological causes (as one biological subset) to schizophrenia.

We certainly agree with the reviewer about the extended MHC (eMHC, chr6 25-34 mb) as the first (2009) and an exceptionally replicable finding in schizophrenia genomics studies of common variation (see our review PMID: 39030273). Consistent with the reviewer’s suggestion, eMHC genes were evaluated in our proteomic work. However, it is standard practice to remove the eMHC region from certain LD-based analyses - and this is specifically recommended by the authors of S-LDSC. The reason for its exclusion is technical and it will skew/bias results if not excluded. This is because the LD structure of the eMHC is singular: perfect correlations between SNPs (phased $r^2 = 1$) over exceptionally long distances (~9 megabases). As we are following standard practices in the field (eg, refs 12 and 38), we believe it best to remove eMHC SNPs from this particular analysis.

- Figure 6 is not helpful but only confusing and should not be shown.

While we can omit it without loss of primary data (all results are represented elsewhere), we believe the schematic provides didactic value, especially for readers less familiar with proteomics. It illustrates that schizophrenia-associated proteomic changes extend across multiple cellular systems—contrasting with the historic synapse-centric view driven by genetics literature. We are happy to: retain this version in the main text, or relocate the full schematic to Supplementary Materials, or remove it entirely. Here, we like to defer to the editor’s judgment.

- the Bast et al 2024 reference is listed several times but is not listed in the references - has this paper been deposited somewhere? If the authors consider data in this paper as essential for the present paper, those results should be accessible one way or another, else there is no need to reference them at all.

We agree. We have now added a proper citation to the manuscript preprint while the Bast et al. paper is currently under review.

- the absence of page numbers in the manuscript is an unnecessary negligence

Apologies, this is an oversight that has been corrected in the updated manuscript.

- Fig S4-S6, S9, S11-S12 – the protein abbreviation codes are too small and not readable, I therefore do not see any sense of presenting this. Either show something readable or do not show it at all.

We agree, these visualizations were not informative and have removed these as suggested. We have ensured that all relevant information that we previously tried to convey in these figures is accessible in the supplementary data tables.

REVIEWER #2

Here the authors detected (quantified?) 36,176 peptides from 6,293 proteins in laminar DPFC dissections, obtained from a cohort consisting of 47 schizophrenia and 49 healthy control subjects. Overall, the study appears to have been well conducted, with a few significant and notable concerns:

- 1. The use of in-gel digestion is no longer standard in the field, having been abandoned almost a decade ago for more robust approaches including FASP and Strap.*

In our experience, in-gel digestion has historically yielded more reliable recovery from human brain tissue than the FASP protocol. We agree with the reviewer that S-trap represents a technical improvement over both in-gels and FASP workflows. Indeed, our lab transitioned to S-trap for most label-free proteomics experiments several years ago. However the dataset reported here was already well underway before that transition was completed, and thus the study necessarily reflects the earlier in-gel protocol.

Large human postmortem proteomics efforts inevitably span multiple years from tissue acquisition through pilot optimization, sample processing, mass-spec acquisition, data analysis, and publication. For example of the long-term efforts even between mass-spec acquisition and publication in the field, the MacDonald et al. manuscript on SCZ proteomic profiling was published online in October 2019 while their Methods section reports “Mass spectrometry data were collected from September 26 through November 4, 2016”. the total turn around time for such projects being easily 4+ years. Our study followed a similar multi-year trajectory that began prior to S-Trap implementation in our lab.

- 2. Cortical laminae are distinguished by both cellular and molecular composition as well as the extent to which they display synaptic alterations in schizophrenia. Thus, the decision to “pool” the layers is not justified, even if similar trends between case and controls are observed in the different layers.*

Thank you for this valuable feedback. We had carefully considered the decision to pool cortical layer data during earlier stages of the study but ultimately opted for a unified data model. We agree with your assessment that independent layer-specific analyses provide a more accurate representation of the molecular and cellular heterogeneity across cortical laminae. Accordingly we statistically re-analyzed our data separately for each layer, updated every main figure and table, and revised the associated Results and Discussion sections throughout the manuscript.

However, the largest issue with this manuscript is that it does not add new findings to the current body of work in the field. Multiple groups (most recently MacDonald 2020, PMID 31642882 and Aryal 2023, PMID: 37171958) have reported alterations in the levels of mitochondrial, synaptic, proteasome, and cytoskeletal proteins in both tissue homogenates AND synaptosome enrichments. The submitted manuscript investigates only tissue homogenates. The proteomic profiling in the latter of these two reports was significantly deeper than the submitted manuscript (711,402 unique peptides that mapped to 8,996 proteins).

We acknowledge the pre-existing, important body of work you refer to and consider our manuscript a novel and significant extension of these findings., Our study contributes a larger set of SCZ-associated proteins and highlights trends of regulation at the level of GO terms.

Following your suggestion we have incorporated the pioneering work by MacDonald and colleagues, and expanded our cross comparison of significant hits across all three major proteomics studies (see updated Fig. 5).

We also provide a more detailed discussion of the commonalities and differences between our study and those by Aryal and MacDonald in the section *Agreement with recent schizophrenia “omics” data*. Key points include:

- MacDonald et al. applied targeted proteins (SRM) to quantify 402 proteins in homogenate and 155 in synaptosome (N = 48 SCZ cases and 48 controls). Our untargeted (and thus unbiased) DIA analysis of homogenates (N = 47 SCZ cases and 49 controls) yields over 6000 protein groups, an order of magnitude more, thereby contributing many additional SCZ-associated proteins of interest to the field.
- Aryal et al. analyzed synaptosomal preparations in great depth (8996 proteins), but please note that their results are orthogonal to our dataset. Schizophrenia pathology extends beyond synaptic dysfunction to include e.g. nuclear processes or non-neuronal responses that synaptosome-focused studies miss. Our broader tissue-level analysis addresses this gap. Thus, studying only synaptosomes as in Aryal et al. shows only part of the landscape, as was also argued for by MacDonald et al. who measured both homogenates and synaptosomes. Therefore, our improved coverage of the SCZ-dysregulated proteome in tissue homogenate has not been performed previously.
- The partial overlap (and same direction of regulation) of significant proteins between all studies (as compared in Figure 5) supports the robustness of our findings. However, beyond this overlap of our data, our study introduces numerous SCZ-associated proteins many of which we extensively discuss in the result section.

As the reviewer pointed out, at the GO level there is much more (though not complete) overlap across the SCZ proteomic studies, which aligns with expectations. For example, the “main pathways” annotated for the protein co-expression networks in SCZ (and Bipolar disorder) as presented in Figure 2 of Aryal et al. contain a few pathways that overlap with our findings, but please note that only few of their results are actually statistically significant. The “Adj-P” column shows as significant: small ribosomes, synaptic structure, copl vesicle, vesicle trafficking and tethering, autophagy. The other

pathways listed in this figure by Aryal et al. are tentative results that did not reach statistical significance, such as “oxidative phosphorylation” which overlaps with our results and is robustly statistically significant in our GO analyses (Bonferroni alpha 0.05).

Analysis of Aryal’s supplementary data table S3 containing control-vs-SCZ GSEA results, reveals GO terms that are significant and up-regulated in both their results and ours including cytoskeleton organization, phosphorylation (p.adj 0.065), kinase binding, autophagy (p.adj 0.075 in our results), GTPase binding. Down-regulated are oxidative phosphorylation, large ribosomes, inner- and outer-mitochondria (we only find inner-). ATP biosynthetic processes are down-regulated and significant in our dataset, with a similar trend in Aryal et al. but not significant (p.adj 0.075). Importantly, as now highlighted in our updated result section *Comparative analysis with existing schizophrenia “omics” data* (page 14-17), our study identifies, beyond the large set of unique SCZ-associated proteins also non-synaptic pathways that are absent from the studies of MacDonald and Aryal, e.g. protein kinase activity, RNA processing, RNA splicing. The latter are absent from Aryal et al. as might be expected due to their study of synapse-enriched biochemical fractions.

Taken together, we have made a large number of manuscript updates following your valuable comments, including many updated and new figures to clearly articulate the novel contributions of our study relative to MacDonald and Aryal. Our work presents (I) a large set of new SCZ-associated proteins not previously identified, (II) several significant pathways not shown by prior proteomic studies, and (III) we also tied some of these findings to single-nucleus RNA sequencing data (performed on the same samples) from a companion paper which adds insight to the cellular heterogeneity across inhibitory neurons, excitatory neurons and non-neuronal cells. All three aspects are novelties not found in the prior studies by MacDonald and Aryal.

Likewise multiple published reports of brain protein quantitative trait loci in the context of psychiatric disease (including schizophrenia) exist, all of which were generated from larger cohorts. Most notably:

1. Luo 2024 (PMID: 38724566) “11,608 proteins across 268. Our analysis revealed 788 cis-acting protein quantitative trait loci associated with the expression of 883 proteins at a genome-wide false discovery rate <5%”. This study includes cases from schizophrenia subjects and significant integration of GWAS and proteomics data in the context of schizophrenia.

2. Wingo 2021 (PMID: 33846625) “we integrated depression genome-wide association study (GWAS) results (N = 500,199) with human brain proteomes (N = 376)”

The main goal and outcome of our study is the comprehensive characterization of SCZ dysregulation in the DLPFC proteome. As discussed in the previous rebuttal point, we think that our dataset and its in-depth analysis and discussion contributes many new SCZ-associated proteins and pathway-levels insights. Additionally, we expanded our scope by including pQTL and LDSC analyses providing orthogonal validation beyond our primary proteomic results and links to prior GWAS studies.

Contrasting our study with Luo et al.;

- Luo et al. used a similar number of SCZ cases (45) as our study (47). The seemingly large difference in total samples size (268 participants) stems from the inclusion of 25 bipolar cases and 198 control so the large discrepancy in N numbers are mostly controls.
 - From their Figure 1: “Postmortem brain samples from a human cohort with 268 participants were used, including 198 normal individuals (CTR), 45 patients with schizophrenia (SCZ), and 25 patients with bipolar (BP).”
- Unlike our study, Luo et al. do not present a control vs schizophrenia comparison using exclusively proteomics data. Instead, their reported SCZ associated proteins/genes are derived from a combination of proteomics and genetics framework with an emphasis on genetic regulation. While this is an interesting and informative approach, we argue that our thorough, unbiased proteomic characterization, along with detailed interpretation at both the protein and pathway levels, provides a unique and valuable contribution to the field.
- It remains very challenging to acquire brain tissue for a large, well controlled cohort of SCZ patients. Expanding such datasets >200 cases is an important future ambition but not realistic at present time as apparent from the N < 50 SCZ cases used in the studies of MacDonald, Aryal, Luo and ours.

The Wingo 2021 study you cited, we noted, focused on depression rather than on schizophrenia. Their reported results on depression pathogenesis do not diminish the impact of novelty of our study on the dysregulation of the DLPFC proteome in schizophrenia patients.

REVIEWER #1

The authors have addressed the main issues raised. Two minor issues remain at this point:

Even though negative, the attempt to identify subsets in the gathered data is important and deserves being mentioned in a paragraph in the main paper with mentioning of the exact statistical/mathematical procedures, rebuttal Figure 1 can go into Supplement.

Figure 6: letters still cannot be distinguished - but I do see the point that the authors make. May I propose to publish Figure 6 as is, but to refer to an additional electronic Supplementary Figure not as part of the Supplementary material but as a small poster-like extra Supplementary Figure so that interested readers are able to decipher the protein abbreviations.

We have added rebuttal Figure 1 to the supplementary information as suggested (now Figure S7). Figure 6 has been moved to supplementary information as well (now Figure S11), as also suggested by the editor, and has been replaced by a vector graphic; readers can download the supplementary information PDF and zoom in to clearly see every protein name/label.